# Open Materials Generation with Stochastic Interpolants

**Philipp Höllmer** [* 1] **Thomas Egg** [* 1] **Maya M. Martirossyan** [* 1] **Eric Fuemmeler** [* 2] **Zeren Shui** [2] **Amit Gupta** [2]
**Pawan Prakash** [3] **Adrian Roitberg** [3] **Mingjie Liu** [3] **George Karypis** [2] **Mark Transtrum** [4] **Richard G. Hennig** [3]
**Ellad B. Tadmor** [2] **Stefano Martiniani** [1]

## Abstract

The discovery of new materials is essential for enabling technological advancements. Computational approaches for predicting novel materials must effectively learn the manifold of stable crystal structures within an infinite design space. We introduce Open Materials Generation (OMatG), a unifying framework for the generative design and discovery of inorganic crystalline materials. OMatG employs stochastic interpolants (SI) to bridge an arbitrary base distribution to the target distribution of inorganic crystals via a broad class of tunable stochastic processes, encompassing both diffusion models and flow matching as special cases. In this work, we adapt the SI framework by integrating an equivariant graph representation of crystal structures and extending it to account for periodic boundary conditions in unit cell representations. Additionally, we couple the SI flow over spatial coordinates and lattice vectors with discrete flow matching for atomic species. We benchmark OMatG's performance on two tasks: Crystal Structure Prediction (CSP) for specified compositions, and *de novo* generation (DNG) aimed at discovering stable, novel, and unique structures. In our ground-up implementation of OMatG, we refine and extend both CSP and DNG metrics compared to previous works. OMatG establishes a new state of the art in generative modeling for materials discovery, outperforming purely flow-based and diffusion-based implementations. These results underscore the importance of designing flexible deep learning frameworks to accelerate progress in materials science. The OMatG code is available at https://github.com/FERMat-ML/OMatG.

## 1. Introduction

A core objective of materials science is the discovery of new synthesizable structures and compounds with the potential to meet critical societal demands. The development of new materials such as room-temperature superconductors (Boeri et al., 2022), high-performance alloys with exceptional mechanical properties (Gludovatz et al., 2014; 2016; George et al., 2019), advanced catalysts (Strmcnik et al., 2016; Nakaya & Furukawa, 2023), and materials for energy storage and generation (Liu et al., 2010; Snyder & Toberer, 2008) holds the potential to drive technological revolutions.

Exploring the vast compositional and structural landscape of multicomponent materials with novel properties is essential, yet exhaustive experimental screening is infeasible (Cantor, 2021). Quantum and classical molecular simulation offer a powerful alternative, enabling a more targeted and efficient exploration. In recent decades, both experimental (Potyrailo et al., 2011; Maier, 2019) and computational (Jain et al., 2011; Curtarolo et al., 2013) high-throughput pipelines have led to a proliferation of materials databases for crystal structures (Bergerhoff et al., 1983; Mehl et al., 2017) and simulations (Blaiszik et al., 2016; Vita et al., 2023; Fuemmeler et al., 2024). These advances have already facilitated the development of more accurate machine-learned interatomic potentials (Batzner et al., 2022; Batatia et al., 2022; Chen & Ong, 2022).

Still, efficiently sampling the manifold of stable materials structures under diverse constraints—such as composition and target properties—remains a major challenge. Traditional approaches to materials discovery have relied on first-principles electronic structure methods such as density functional theory (DFT)[1]—or more sophisticated theory, depending on the property (Booth et al., 2013; Zaki et al., 2014; Isaacs & Marianetti, 2020)—which, while powerful and fairly accurate, are very computationally expensive. These methods include *ab initio* random structure searching (AIRSS) (Pickard & Needs, 2011) or genetic algorithms for structure and phase prediction (Tipton & Hennig, 2013), both of which have successfully predicted new crystal struc-

---

*Equal contribution [1]New York University [2]University of Minnesota [3]University of Florida [4]Brigham Young University. Correspondence to: Stefano Martiniani <sm7683@nyu.edu>.

*Proceedings of the 42nd International Conference on Machine Learning*, Vancouver, Canada. PMLR 267, 2025. Copyright 2025 by the author(s).

---

[1]See Appendix A for a list of acronyms used throughout this paper.

tures and some of which have even been experimentally realized (Oganov et al., 2019). However, the high computational cost of these approaches has limited the scope and speed of material exploration, highlighting the need for cutting-edge ML techniques to significantly accelerate the discovery of stable inorganic crystalline materials.

## 1.1. Related Works

Recent advances in machine learning techniques have generated significant interest in applying data-driven approaches for inorganic materials discovery. Among these, Graph Networks for Materials Exploration (GNoME) has demonstrated remarkable success by coupling coarse sampling strategies for structure and composition with AIRSS that leverages a highly accurate machine-learned interatomic potential (MLIP) to predict material stability, leading to the identification of millions of new candidate crystal structures (Merchant et al., 2023). Other frameworks have approached the generation of composition and structure jointly through fully ML-based methods. Crystal Diffusion Variational Autoencoder (CDVAE) leverages variational autoencoders and a graph neural network representation to sample new crystal structures from a learned latent space (Xie et al., 2022). To date, state-of-the-art performance in both crystal structure prediction for given compositions and *de novo* generation of novel stable materials has been achieved by diffusion models such as DiffCSP (Jiao et al., 2023) and MatterGen (Zeni et al., 2025), as well as conditional flow-matching frameworks such as FlowMM (Miller et al., 2024).

While these approaches have demonstrated that ML can push the boundaries of computational materials discovery, it remains uncertain whether score-based diffusion or flow-matching represents the definitive methodological frameworks for this problem. Furthermore, the extent to which the optimal approach depends on the training data remains an open question. Thus far, each new method has typically outperformed its predecessors across datasets.

## 1.2. Our Contribution

The work we present in this paper is the first implementation and extension of the stochastic interpolants (SIs) framework (Albergo et al., 2023) for the modeling and generation of inorganic crystalline materials. SIs are a unifying framework that encompasses both flow-matching and diffusion-based methods as specific instances, while offering a more general and flexible framework for generative modeling. In this context, SIs define a stochastic process that interpolates between pairs of samples from a known base distribution and a target distribution of inorganic crystals. By learning the velocity term of an ordinary differential equation (ODE) or the drift term of a stochastic differential equations (SDE), new samples can be generated by numerically integrating

these equations. The flexibility of the SI framework stems from the ability to tailor the choice of interpolants, and the incorporation of an additional random latent variable, further enhancing its expressivity. With their rich parameterization, SIs thus provide an ideal framework for optimizing generative models for materials design.

We implement the SI framework in the open-source Open Materials Generation (OMatG) package, released alongside this paper. OMatG allows to train and benchmark models for two materials generation tasks: *Crystal structure prediction* (CSP) which only learns to generate atomic positions and lattice vectors for a given composition, and *de novo generation* (DNG) which learns to generate both crystal structure and composition to predict novel materials. We discover that optimizing interpolation schemes for different degrees of freedom of the crystal unit cell substantially improves performance across diverse datasets. As a result, our approach achieves a new state of the art—outperforming both DiffCSP (Jiao et al., 2023) and FlowMM (Miller et al., 2024) in CSP and DNG, as well as MatterGen (Zeni et al., 2025) in DNG—across all evaluated datasets under existing, revised, and new performance measures.

## 2. Background

### 2.1. Diffusion Models

A widely used approach in generative modeling uses diffusion models (Sohl-Dickstein et al., 2015), which define a stochastic process that progressively transforms structured data into noise *via* a predefined diffusion dynamic. A model is then trained to approximate the reverse process, enabling the generation of new samples, typically by integrating a corresponding SDE.

Score-based diffusion models (SBDMs) are an instantiation of diffusion models that learn a score function—the gradient of the log probability density—to guide the reversal of the diffusion process via numerical integration (Song et al., 2021). SBDMs have demonstrated remarkable success in generating high-quality and novel samples across a wide range of applications where the target distribution is complex and intractable, such as photorealistic image generation (Saharia et al., 2022) and molecular conformation prediction (Corso et al., 2023).

### 2.2. Conditional Flow Matching

Conditional flow matching (CFM) (Liu, 2022; Lipman et al., 2023; Albergo & Vanden-Eijnden, 2023) is a generative modeling technique that learns a flow which transports samples from a base distribution at time $t = 0$ to a target distribution at time $t = 1$. This process defines a probability path that describes how samples are distributed at any intermediate time $t \in [0, 1]$. The velocity field associated with

this flow governs how individual samples evolve over time. CFM learns the velocity indirectly by constructing conditional vector fields that are known *a priori*. Once trained, samples drawn from the base distribution can be evolved numerically to generate new samples from the target distribution. Originally, CFM was formulated using Gaussian conditional probability paths, but Tong et al. (2024) later extended this framework to allow for arbitrary probability paths and couplings between base and target distributions. A further extension, particularly relevant to physics and chemistry, is Riemannian flow matching (RFM), which generalizes CFM to Riemannian manifolds (Chen & Lipman, 2024). This allows in particular to use the flow-matching framework for systems with periodic boundary conditions as they appear in unit cell representations of inorganic crystals (Miller et al., 2024).

# 3. Open Materials Generation

## 3.1. Stochastic Interpolants

SIs provide a unifying mathematical framework for generative modeling, generalizing both SBDMs and CFM (Albergo et al., 2023). The SI $x(t, x_0, x_1, z)$ bridges the base distribution $\rho_0$ with a target distribution $\rho_1$ by learning a time-dependent map. In this work, we focus on stochastic interpolants of the form:

$$x_t \equiv x(t, x_0, x_1, z) = \alpha(t)x_0 + \beta(t)x_1 + \gamma(t)z. \quad (1)$$

Here, $t \in [0, 1]$ represents time and $(x_0, x_1)$ are paired samples drawn from $\rho_0$ and $\rho_1$, respectively. The random variable $z$ is drawn from a standard Gaussian $\mathcal{N}(0, \boldsymbol{I})$ independently of $x_0$ and $x_1$. The functional forms of $\alpha$, $\beta$, and $\gamma$ are flexible, subject to few constraints (see Appendix B.2). The inclusion of the latent variable $\gamma(t)z$ allows sampling of an ensemble of paths around the mean interpolant $I(t, x) = \alpha(t)x_0 + \beta(t)x_1$, and is theorized to improve generative modeling by promoting smoother and more regular learned flows (Albergo et al., 2023).

The time-dependent density $\rho_t$ of the stochastic process $x_t$ in Eq. (1) can be realized either *via* deterministic sampling through an ODE (derived from a transport equation) or stochastic sampling through an SDE (derived from a Fokker–Planck equation) only requiring $x_0 \sim \rho_0$ (see Appendix B.1). This enables generative modeling by evolving samples from a known base distribution $\rho_0$ to the target distribution $\rho_1$. For both ODE- and SDE-based sampling, the required velocity term $b^\theta(t, x)$ is learned by minimizing the loss function

$$\mathcal{L}_b(\theta) = \mathbb{E}_{t,z,x_0,x_1} \big[ |b^\theta(t, x_t)|^2 \\ - 2 \, \partial_t x(t, x_0, x_1, z) \cdot b^\theta(t, x_t) \big], \quad (2)$$

where the expectation is taken independently over $t \sim \mathcal{U}(0, 1)$, the uniform distribution between 0 and 1, $z \sim$

$\mathcal{N}(0, \boldsymbol{I})$, $x_0 \sim \rho_0$, and $x_1 \sim \rho_1$. For SDE-based sampling, an additional denoiser $z^\theta(t, x)$ must be learned by minimizing an additional loss

$$\mathcal{L}_z^\theta(\theta) = \mathbb{E}_{t,z,x_0,x_1} \big[ |z^\theta(t, x_t)|^2 - 2 \, z^\theta(t, x_t) \cdot z \big]. \quad (3)$$

The velocity term, along with the denoiser in the case of SDE-based sampling, enables the generation of samples from the target distribution (Albergo et al., 2023). Note that minimizing with respect to these loss functions amounts to minimizing with respect to a mean-squared error loss function (see Appendix C.2). For ODE-based sampling, $\gamma(t) = 0$ in the interpolant $x(t, x_0, x_1, z)$ is a possible choice. However, for SDE-based sampling, $\gamma(t) > 0$ is required for all $t \in (0, 1)$ (see Appendix B.1).

By appropriately selecting interpolation functions $\alpha$, $\beta$, $\gamma$ and choosing between deterministic (ODE) and stochastic (SDE) sampling schemes (see Appendix B.2), the SI framework not only recovers CFM and SBDM as special cases (see Appendix B.9) but also enables the design of a broad class of novel generative models. The strength of OMatG's SI implementation for materials discovery lies in its ability to tune both the interpolation and sampling schemes, as illustrated in Fig. 1 for a pair of structures sampled from $\rho_0$ and $\rho_1$. By systematically optimizing over this large design space, we achieve superior performance for CSP and DNG tasks across datasets, as discussed in Section 5.

## 3.2. Crystal Representation and Generation

A crystalline material is defined by its idealized repeat unit, or unit cell, which encodes its periodicity. In the OMatG representation, a unit cell is described by separating the material's chemical composition—given by its atomic species $\boldsymbol{A} \in \mathbb{Z}_{>0}^N$, where $N$ is the number of atoms in the unit cell—from its structural representation—its fractional coordinates $\boldsymbol{X} \in [0, 1)^{3 \times N}$ with periodic boundaries and lattice vectors $\boldsymbol{L} \in \mathbb{R}^{3 \times 3}$. During training, all three components $\{\boldsymbol{A}, \boldsymbol{X}, \boldsymbol{L}\}$ are considered simultaneously. We apply the SI framework only to the continuous structural representations $\{\boldsymbol{X}, \boldsymbol{L}\}$ with loss functions defined in Eqs (2) and (3), and use discrete flow matching (DFM) on the chemical species $\boldsymbol{A}$ (see Section 3.2.3) (Gat et al., 2024). The number of atoms $N$ in the structure $x_0$ sampled from the base distribution $\rho_0$ is determined by the number of atoms in the corresponding structure $x_1$ sampled from the target distribution $\rho_1$.

### 3.2.1. ATOMIC COORDINATES

For treating fractional coordinates, we implement a variety of periodic interpolants that connect the base and target data distributions (see Section 4.1). We specify the base distribution for the fractional coordinates $\boldsymbol{x} \in [0, 1)$ for all $\boldsymbol{x} \in \boldsymbol{X}$ *via* a uniform distribution (except for the score-based dif-

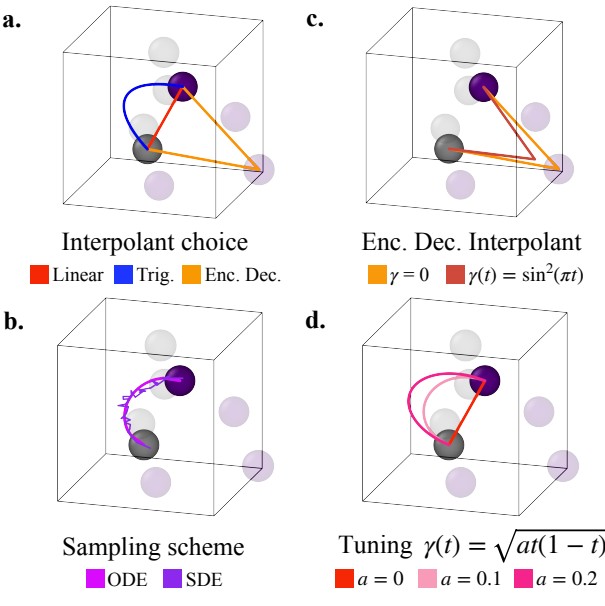

*Figure 1.* Visualization of the tunable components of the SI framework for bridging samples $x_0$ (gray particles) and $x_1$ (purple particles). Interpolation paths are shown only for one pair of highlighted particles. **(a)** The choice of the interpolant changes the path of the time-dependent interpolation trajectory. **(b)** During inference, the learned velocity term $b^\theta(t, x)$ and denoiser $z^\theta(t, x)$ generate new samples *via* ODE or SDE integration, here for a linear interpolant with $\gamma = \sqrt{0.07t(1-t)}$. **(c)** The inclusion of a latent variable $\gamma(t)z$ changes the interpolation path. For SDE-based sampling, $\gamma(t) > 0$ is required. **(d)** The function $\gamma(t, a) = \sqrt{at(1-t)}$ depends on $a$ that also influences the interpolation path.

fusion interpolant that, following the approach of Jiao et al. (2023), uses a wrapped normal distribution $\rho_0(\boldsymbol{x})$ which becomes a uniform distribution in the limit of large variance). To satisfy periodic boundary conditions on the paths defined by the interpolants, we extend the SI framework to the surface of a four-dimensional torus in this paper. Reminiscent of RFM (Chen & Lipman, 2024), the linear interpolant on the torus traverses a path equivalent to the shortest-path geodesic which is always well-defined.[2] Other interpolants, however, are more complex. In order to uniquely define them, we always define the interpolation with respect to the shortest-path geodesic. That is, for interpolation between $\boldsymbol{x}_0$ and $\boldsymbol{x}_1$ with a periodic boundary at 0 and 1, we first unwrap $\boldsymbol{x}_1$ to the periodic image $\boldsymbol{x}_1'$ which has the shortest possible distance from $\boldsymbol{x}_0$. Following this, the interpolation between $\boldsymbol{x}_0$ and $\boldsymbol{x}_1'$ is computed given a choice of interpolant, and the traversed path is wrapped back into the boundary from 0 to 1. This approach is illustrated in Appendix B.5.

---

[2]The only exception being when two points are precisely half the box length apart. However, this case is not relevant for the given base distribution.

### 3.2.2. LATTICE VECTORS

Lattice vectors $\boldsymbol{L}$ are treated with a wide range of (non-periodic) stochastic interpolants (see Section 4.1 again). To construct the base distribution, we follow Miller et al. (2024) and construct an informative base distribution $\rho_0(\boldsymbol{L})$ by combining a uniform distribution over the lattice angles with a log-normal distribution fitted to the empirical distribution of the lattice lengths in each target dataset. This choice brings the base distribution closer to the target distribution. Unlike SBDM, which requires a Gaussian base distribution, the SI framework allows such flexibility. Importantly, the model still has to learn to generate a joint, correlated distribution of lattice vectors, fractional coordinates, and atomic species.

### 3.2.3. ATOMIC SPECIES

The discrete nature of chemical compositions $\boldsymbol{A}$ in atomic crystals requires a specialized approach for generative modeling. To address this, we implement discrete flow matching (DFM) (Campbell et al., 2024). In our implementation of the DFM framework, each atomic species $\boldsymbol{a} \in \boldsymbol{A}$ can take values in $\{1, 2, \ldots, 100\} \cup \{M\}$; where $\{1 - 100\}$ are atomic element numbers and $M$ is a masking token used during training. The base distribution is defined as $\rho_0(\boldsymbol{a}) = [M]^N$, meaning that initially all $N$ atoms are masked. As sampling progresses, the identities of the atoms evolve *via* a continuous-time Markov Chain (CTMC), and are progressively unmasked to reveal valid atomic species. At $t = 1$, all masked tokens are replaced. To learn this process, we define a conditional flow $p_{t|1}(\boldsymbol{a}_t|\boldsymbol{a}_1)$ that linearly interpolates in time from the fully masked state $\boldsymbol{a}_0$ toward $\boldsymbol{a}_1$ and thus yields the composition $\boldsymbol{a}_t$ of the interpolated structure $x_t$. Based on these conditional flows, a neural network is trained to approximate the denoising distribution $p_{1|t}^\theta(\boldsymbol{a}_1|x_t)$, which yields the probability for the composition $\boldsymbol{a}_1$ given the entire structure $x_t$, by minimizing a cross-entropy loss

$$\mathcal{L}_{\text{DFM}}(\theta) = \mathbb{E}_{t,x_1,x_t}\left[\log p_{1|t}^\theta(\boldsymbol{a}_1|x_t)\right]. \quad (4)$$

In doing this, we are able to directly construct the marginal rate matrix $R_t^\theta(\boldsymbol{a}_t, i)$ for the CTMC that dictates the rate of $\boldsymbol{a}_t$ at time $t$ jumping to a different state $i$ during generation (see Appendix B.6). It is important to note that the learned probability path is a function of the entire atomic configuration $\{\boldsymbol{A}, \boldsymbol{X}, \boldsymbol{L}\}$ which is necessary for the prediction of chemical composition from structure.

### 3.3. Joint Generation with Stochastic Interpolants

For both CSP and DNG tasks, we seek to generate samples from a joint distribution over multiple coordinates. For DNG, this joint distribution $\rho_1$ encompasses all elements of a crystal unit cell. For CSP we similarly model the joint

distribution, $\rho_1$, but with atom types fixed to compositions sampled from the target dataset. For both tasks, the total loss function is formulated as a weighted sum of the individual loss functions for each variable (see Appendix C.2), and their relative weights are optimized (see Appendix C.3). We illustrate both types of models and their structure generation process in Fig. 2a.

Additionally, for DNG, we consider a two-step process in which composition is learned separately from structure, as seen in Fig. 2b. In this approach, we first train a chemical formula prediction (CFP) model (see Appendix C.1) to generate compositions optimized for SMACT stability (Davies et al., 2019), similarity in the distribution of $N$-arity of known structures, as well as uniqueness and novelty. The predicted compositions are then used as input for a pretrained CSP model, which generates the corresponding atomic configurations.

## 4. Methodology

### 4.1. Choice of Interpolant

In training OMatG, we optimize the choice of the interpolating function that is used during training for the lattice vectors (without periodic boundary conditions) and the fractional coordinates (with periodic boundary conditions). We consider four interpolants of the form defined in Eq. (1), each shaping the interpolation trajectory differently (see also Appendices B.2 and B.9 for further details).

The **linear** interpolant defines a constant velocity trajectory from $x_0$ to $x_1$. When combined with an ODE sampling scheme and $\gamma = 0$, this reproduces the specific instantiation of CFM implemented in FlowMM (see Appendix B.9). However, combining the linear interpolant with an SDE sampling scheme or nonzero $\gamma$ already introduces key differences. The inclusion of the latent variable can promote smoother learned flows (Albergo et al., 2023), while stochastic sampling alters the generative dynamics compared to the deterministic formulation in FlowMM. The **trigonometric** interpolant prescribes trajectories with more curvature than the linear interpolant. The **encoder-decoder** interpolant first evolves samples from $\rho_0$ at $t = 0$ to follow an intermediate Gaussian distribution at a switch time $T_{\text{switch}}$, before mapping them to samples from the target distribution $\rho_1$ at $t = 1$. This approach has been found to interpolate more smoothly between distributions, potentially mitigating the formation of spurious features in the probability path at intermediate times (Albergo et al., 2023). Lastly, we consider **variance-preserving score-based diffusion** (VP SBD) and **variance-exploding score-based diffusion** (VE SBD) interpolants. When paired with an SDE sampling scheme, these interpolants are mathematically equivalent to the corresponding SBDM, but on the continuous time

interval $[0, 1]$. Different noise schedules of the variance-preserving and variance-exploding SBDMs can likewise be encoded in different variants of the VP and VE SBD interpolants (see Appendix B.9). For the results presented in this paper, we only consider a constant noise schedule for the VP SBD interpolant, and a geometric noise schedule for the VE SBD interpolant. The SBD interpolants assume that $\rho_0$ is a Gaussian distribution, and unlike the previous interpolants it involves no explicit latent variable; instead the $\alpha(t)x_0$ term takes on this role. The incorporation of VP and VE SBD interpolants enables OMatG to reproduce similar conditions to those in DiffCSP and MatterGen.

The trajectory of the encoder-decoder interpolant between times $t = T_{\text{switch}}$ and $t = 1$ resembles that of the SBD interpolants between times $t = 0$ and $t = 1$. For the example of using the encoder-decoder interpolant only for the coordinates, however, we emphasize that the Gaussian-distributed coordinates at $t = T_{\text{switch}}$ are conditioned on other coordinates that are partially interpolated at this point. Conversely, for SBD interpolation, the Gaussian distributed coordinates at $t = 0$ are only conditioned on other random variables since, at this point, all elements of $x_0$ are randomly distributed.

To investigate how different interpolants affect generative performance, we consider all interpolants outlined above for both the atomic positions $\boldsymbol{X}$ and the lattice vectors $\boldsymbol{L}$. We noted that learning accurate velocities and denoisers for the atomic positions, $\boldsymbol{X}$ is more challenging than for the other degrees of freedom. Accordingly, we optimize all hyperparameters—including the choice of interpolant for $\boldsymbol{L}$—separately for each interpolant applied to $\boldsymbol{X}$. This results in a set of experiments specific to the positional interpolants, where the best performing lattice interpolant may vary.

### 4.2. Equivariant Representation of Crystal Structures

Imposing inductive biases on the latent representation of the crystal structure can promote data efficiency and improve learning. The CSPNet architecture (Jiao et al., 2023), originally adopted in DiffCSP, is an equivariant graph neural network (EGNN) (Satorras et al., 2021) that produces a permutation- and rotation-equivariant, as well as translation-invariant representation of the crystal structures.

In the current OMatG implementation, we employ CSPNet as an encoder that is trained from scratch. The CSPNet architecture encodes atomic types using learnable atomic embeddings and represents fractional coordinates through sinusoidal positional encodings (see Appendix C.1). These features are processed through six layers of message-passing, after which the encoder produces the velocity $b^\theta(t, x)$ of both the lattice and the fractional coordinates, as well as potentially predicting the denoiser $z^\theta(t, x)$. For DNG, the

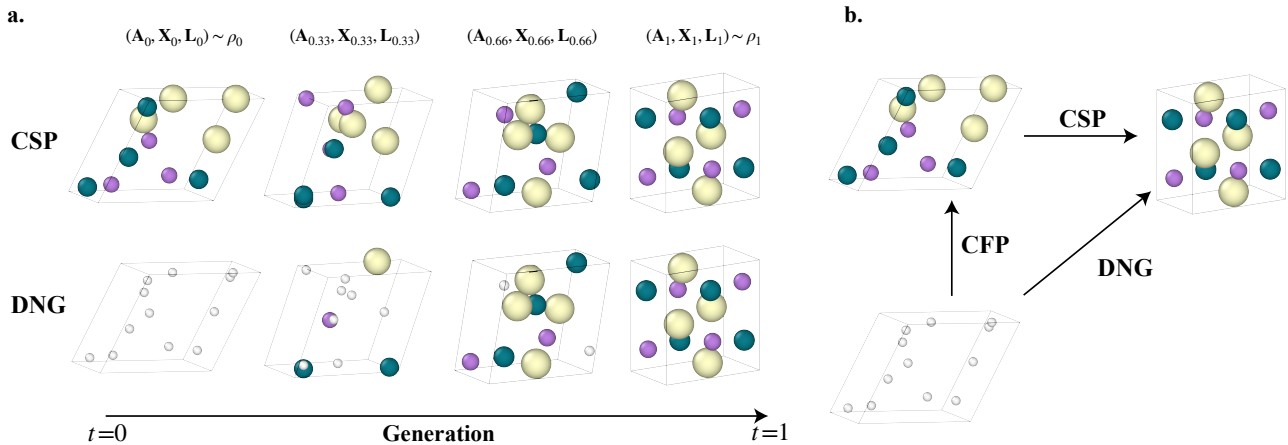

*Figure 2.* Illustration of crystal structure prediction (CSP) and *de novo* generation (DNG) tasks. (**a**) For CSP, the species $\boldsymbol{A}$ are fixed with known compositions from $t = 0$. From this, we predict $\boldsymbol{X}$ and $\boldsymbol{L}$ from randomly sampled initial values. For DNG, we predict $(\boldsymbol{A}, \boldsymbol{X}, \boldsymbol{L})$ jointly. Our implementation of discrete flow matching (DFM) initializes $\boldsymbol{A}$ as a sequence of masked particles that are unmasked through a series of discrete jumps to reveal a physically reasonable composition. (**b**) Two avenues for performing DNG of materials. The first uses two steps: a chemical formula prediction (CFP) model predicts compositions and then uses a CSP model to find accompanying stable structures. The second trains a DNG model over cell, species, and fractional coordinates jointly as shown in (a).

network must also predict $\log p_{1|t}^{\theta}(\boldsymbol{a}_1|x_t)$. The resulting outputs inherently preserve the permutation, rotational, and translational symmetries embedded in CSPNet.

The output of CSPNet is invariant with respect to translations of the fractional coordinates in the input. Thus, one should, in principle, use a representation of the fractional coordinates that does not contain any information about translations. While this is straightforward in Euclidean space by removing the mean of the coordinates of the given structure, this cannot be done with periodic boundary conditions where the mean is not uniquely defined. We follow Miller et al. (2024) and instead remove the well-defined center-of-mass motion when computing the ground-truth velocity $\partial_t x(t, x_0, x_1, z)$ in Eq. (2).

Alternative EGNNs such as NequIP (Batzner et al., 2022), M3GNet (Chen & Ong, 2022), or MACE (Batatia et al., 2022) which have been widely used for the development of MLIPs can also serve as plug-and-play encoders within OMatG's SI framework. Integrating different architectures is a direction that we plan to explore in future iterations of the framework.

### 4.3. Comparison to other Frameworks

We compare our results to DiffCSP and FlowMM models for both CSP and DNG. For DNG, we additionally consider the MatterGen-MP model that was trained on the same *MP-20* dataset as OMatG's DNG models. We detail in Section 5 how we improve the extant benchmarks used in the field and therefore recompute all CSP and DNG benchmarks for

these models. In nearly all cases, we were able to generate structures using the DiffCSP, FlowMM, and MatterGen source codes whose metrics closely matched the previously reported metrics in their respective manuscripts. For Diff-CSP and MatterGen, we relied on published checkpoints while we retrained FlowMM from scratch. The observed differences can be attributed to the use of a newer version of SMACT composition rules[3] (Davies et al., 2019) and to natural fluctuations during generation and model retraining.

Since the focus of this work is to assess our model's ability to learn unconstrained and unconditioned flows, we do not compare against symmetry-constrained generation methods (AI4Science et al., 2023; Cao et al., 2024; Zhu et al., 2024; Kazeev et al., 2024; Jiao et al., 2024). Symmetry constraints can be incorporated in future extensions of the flexible OMatG framework.

## 5. Experiments

### 5.1. Performance Metrics

**Crystal structure prediction.** We assess the performance of OMatG's and competing models using a variety of standard (introduced in (Xie et al., 2022; Zeni et al., 2025)), refined, and contributed benchmarks. For the CSP task, we generate a structure for every composition in the test dataset. We then attempt to match every generated structure with the corresponding test structure using Pymatgen's

---

[3]The SMACT Python library updated its default oxidation states with the release of version 3.0.

*Table 1.* Results from crystal structure prediction. Match rate and RMSE of matched structures without (left) and with (right) filtering for structural and compositional validity are reported for all models. For OMatG's models, the choice of positional interpolant, latent variable component $\gamma$, and sampling scheme are noted. For all SDE sampling schemes the inclusion of $\gamma$ is assumed and not noted; for SBD interpolants $\gamma$ is not relevant. Further details and complete results for *perov-5* and *MP-20* can be found in Appendix C.3.

| Method | *perov-5* | | *MP-20* | | *MPTS-52* | | *Alex-MP-20* | |
|---|---|---|---|---|---|---|---|---|
| | Match (%) ↑ | RMSE ↓ | Match (%) ↑ | RMSE ↓ | Match (%) ↑ | RMSE ↓ | Match (%) ↑ | RMSE ↓ |
| DiffCSP | 53.08 / 51.94 | 0.0774 / 0.0775 | 57.82 / 52.51 | **0.0627 / 0.0600** | 15.79 / 14.29 | **0.1533** / 0.1489 | - | |
| FlowMM | 53.63 / 51.86 | 0.1025 / 0.0994 | 66.22 / 59.98 | 0.0661 / 0.0629 | 22.29 / 20.28 | 0.1541 / **0.1486** | - | |
| OMatG | | | | | | | | |
| Linear (ODE) w/o $\gamma$ | 51.86 / 50.62 | **0.0757 / 0.0760** | **69.83 / 63.75** | 0.0741 / 0.0720 | **27.38 / 25.15** | 0.1970 / 0.1931 | 72.02 / 64.23 | 0.0683 / 0.0671 |
| Linear (SDE) w/ $\gamma$ | **74.16 / 72.87** | 0.3307 / 0.3315 | 68.20 / 61.88 | 0.1632 / 0.1611 | 23.95 / 21.70 | 0.2402 / 0.2353 | 61.07 / 54.45 | 0.1870 / 0.1860 |
| Trig (SDE) w/ $\gamma$ | 73.37 / 71.60 | 0.3610 / 0.3614 | **68.90 / 62.65** | 0.1249 / 0.1235 | 24.51 / 22.26 | 0.1867 / 0.1804 | **72.50 / 64.71** | 0.1261 / 0.1251 |
| Enc-Dec (ODE) w/ $\gamma$ | **68.08 / 64.60** | 0.4005 / 0.4003 | 55.15 / 49.45 | 0.1306 / 0.1260 | 14.65 / 13.53 | 0.2543 / 0.2500 | 68.11 / 60.58 | 0.0957 / 0.0938 |
| VP SBD (ODE) | **83.06 / 81.27** | 0.3753 / 0.3755 | 45.57 / 39.48 | 0.1880 / 0.1775 | 9.66 / 8.36 | 0.3088 / 0.3041 | 46.23 / 39.96 | 0.1718 / 0.1618 |
| VE SBD (ODE) | **60.18 / 52.97** | 0.2510 / 0.2337 | 63.79 / 57.82 | 0.0809 / 0.0780 | 21.42 / 19.57 | 0.1740 / 0.1702 | 67.79 / 60.25 | **0.0674 / 0.0649** |

*Table 2.* Results from *de novo* generation of 10 000 structures with models trained on the *MP-20* dataset. The integration steps for OMatG is chosen based on best overall performance. For OMatG's model, the choice of positional interpolant, latent variable component $\gamma$, and sampling scheme are noted. Best scores in each category are bolded.[*]

| Method | Integration steps | Validity (% ↑) | | | Coverage (% ↑) | | Property (↓) | | |
|---|---|---|---|---|---|---|---|---|---|
| | | Structural | Composition | Combined | Recall | Precision | wdist ($\rho$) | wdist ($N$ary) | wdist ($\langle CN \rangle$) |
| DiffCSP | 1000 | 99.91 | 82.68 | 82.65 | **99.67** | 99.63 | 0.3133 | 0.3193 | 0.3053 |
| FlowMM | 1000 | 92.26 | 83.11 | 76.94 | 99.34 | 99.02 | 1.0712 | 0.1130 | 0.4405 |
| MatterGen-MP | 1000 | 99.93 | 83.89 | 83.89 | 96.62 | **99.90** | 0.2741 | 0.1632 | 0.4155 |
| OMatG | | | | | | | | | |
| Linear (SDE) w/ $\gamma$ | 710 | 99.04 | 83.40 | 83.40 | 99.47 | 98.81 | 0.2583 | **0.0418** | 0.4066 |
| Trig (ODE) w/ $\gamma$ | 680 | 95.05 | 82.84 | 82.84 | 99.33 | 94.75 | **0.0607** | **0.0172** | **0.1650** |
| Enc-Dec (ODE) w/ $\gamma$ | 840 | 97.25 | **86.35** | **84.19** | 99.62 | 99.61 | 0.1155 | **0.0553** | **0.0465** |
| VP SBD (SDE) | 870 | 93.38 | 80.66 | 80.66 | 98.95 | 92.76 | 0.1865 | **0.0768** | **0.1637** |
| CFP + CSP [Linear (ODE) w/o $\gamma$] | 130+210 | 97.95 | 79.68 | 78.21 | **99.67** | 99.50 | 0.5614 | 0.2008 | 0.6256 |

[*]We do not bold any values in the structural validity category as the CDVAE model reports the state of the art with 100% structural validity. For the Wasserstein distances of the density and $N$ary distributions, we only bold values lower than 0.075 and 0.079 respectively, as these were the values reported by FlowMM for their model with 500 integration steps (not included in this table).

StructureMatcher module (Ong et al., 2013) with tolerances (stol = 0.5, ltol = 0.3, angletol = 10). We finally report the match rate and the average root-mean square displacement (RMSE) between the test structures and *matched* generated structures. Here, the RMSEs computed by Pymatgen are normalized by $(V/N)^{1/3}$, where $V$ is the (matched) volume and $N$ is the number of atoms. During hyperparameter optimization, we only attempt to maximize the match rate (see Appendix C.3).

Previously reported match rates filtered the matched generated structures by their structural and compositional validity (see Appendix D.2). We note, however, that the datasets themselves contain invalid structures—for example, the *MP-20* test dataset has $\sim 10\%$ compositionally invalid structures. Thus, we argue that the removal of these invalid structures for computation of match rate and RMSE is not reasonable for assessing learning performance; we do, however, provide both match rates (with and without validation filtering).

***De novo* generation.** For the DNG task, metrics include validity (structural and compositional), coverage (recall and precision), and Wasserstein distances between distributions of properties including density $\rho$, number of unique elements $N$ (*i.e.*, an $N$ary material), and average coordination number by structure $\langle CN \rangle$. We newly introduce the average coordination number benchmark due to the difficulty of generating symmetric structures; a structure's average coordination number is a useful fingerprint, and higher-coordinated structures tend to be more symmetric.

The previous DNG metrics are used in conjunction during optimization of hyperparameters (see Appendix C.3). For the best models, we then structurally relax the generated structures in order to calculate the stability and the S.U.N. (stable, unique, and novel) rates using MatterGen's code base (MatterGen, 2025). The S.U.N. rate is defined as the percentage of generated structures that are *stable* with respect to a reference convex hull (within 0.1 eV/atom), are not found within the reference set (*novel*), and are not dupli-

cated within the generated set itself (*unique*). The machine-learned interatomic potential MatterSim (Yang et al., 2024) is utilized for initial structural relaxation, which is subsequently followed by a more computationally expensive DFT relaxation. Full workflow details are given in Appendix D.4. In addition to the stability-based metrics, we also report the root mean squared displacement (RMSD) between the generated and relaxed structures (unnormalized, in units of Å).

## 5.2. Benchmarks and Datasets

We use the following datasets to benchmark the models: *perov-5* (Castelli et al., 2012), a dataset of perovskites with $18\,928$ samples with five atoms per unit cell in which only lattice lengths and atomic types change; *MP-20* (Jain et al., 2013; Xie et al., 2022) from the Materials Project that contains $45\,231$ structures with a maximum of $N = 20$ atoms per unit cell, and *MPTS-52* (Baird et al., 2024) which is a chronological data split of the Materials Project with $40\,476$ structures with up to $N = 52$ atoms per unit cell and is typically the most difficult to learn. We use the same 60-20-20 splits as Xie et al. (2022); Jiao et al. (2023); Miller et al. (2024). Additionally, we consider the *Alex-MP-20* dataset (Zeni et al., 2025), where we used an 80-10-10 split constructed from MatterGen's 90-10 split, in which we removed 10% of the training data to create a test dataset. This dataset contains $675\,204$ structures with 20 or fewer atoms per unit cell from the *Alexandria* (Schmidt et al., 2022a;b) and *MP-20* datasets. We do not include the *carbon-24* dataset (Pickard, 2020) in our results, as the current match rate metric is ill-defined for this dataset; because all elements are carbon, it is not clear how many generated structures are unique and producing a structure that matches one in the reference dataset is trivial.[4]

## 5.3. Results

**Crystal structure prediction.** We report the CSP performance of DiffCSP, FlowMM, and six OMatG models on the four benchmark datasets in Tab. 1. Further ablation results across OMatG variants can be found in Tables 6 and 8 in the Appendix. OMatG significantly outperforms previous approaches with respect to match rate on all datasets. We highlight the strong match rates on the *perov-5* dataset achieved using the VP SBD and trigonometric positional interpolants with ODE sampling schemes, as shown in Table 1 and Table 6, that greatly surpass (by a factor of 1.6) the match rates of previous models. We note that the relative performance for ODE *vs.* SDE sampling schemes depends on the positional interpolant. OMatG also outperforms pre-

---

[4]Previous papers (Xie et al., 2022; Jiao et al., 2023; Miller et al., 2024) report match rate for *carbon-24*, but they do not compare each generated structure to the entirety of the reference dataset; their results suggest the match tolerance is larger than the differences between the *carbon-24* structures themselves.

*Table 3.* Stability (defined as $\leq 0.1$ eV/atom above hull), uniqueness, and novelty results from *de novo* generation on the *MP-20* dataset computed for the same models as in Tab. 2. All evaluations are calculated with respect to 1000 relaxed structures (see Appendix D.4), utilizing the MatterGen code base (MatterGen, 2025) and the included reference *Alex-MP*-20 dataset. The average RMSD is between the generated and the relaxed structures, and the average energy above hull is reported in units of eV/atom.

| Method | $\langle E \rangle / N$ ($\downarrow$) above hull | RMSD (Å, $\downarrow$) | Novelty Rate (%, $\uparrow$) | Stability Rate (%, $\uparrow$) | S.U.N. Rate (%, $\uparrow$) |
|---|---|---|---|---|---|
| DiffCSP | 0.1751 | 0.3861 | 70.04 | 43.43 | 15.95 |
| FlowMM | 0.1917 | 0.6316 | 69.13 | 41.20 | 11.73 |
| MatterGen-MP | 0.1772 | **0.1038** | 72.40 | 44.79 | 20.30 |
| OMatG | | | | | |
| Linear (SDE) w/ $\gamma$ | 0.1808 | 0.6357 | **73.31** | **46.18** | **22.48** |
| Trig (ODE) w/ $\gamma$ | **0.1670** | 0.6877 | 66.45 | **52.81** | 21.10 |
| Enc-Dec (ODE) w/ $\gamma$ | **0.1482** | 0.4187 | 55.21 | **60.04** | 18.77 |
| VP SBD (SDE) | 0.2120 | 0.7851 | **76.25** | 40.83 | **20.73** |
| CFP + CSP | 0.2302 | 0.5375 | **79.08** | 39.38 | 20.17 |

vious models' match rate for CSP on the *MP-20* and *MPTS-52* datasets with the linear (both ODE and SDE sampling scheme) and the trigonometric positional interpolants. Finally, OMatG establishes the first performance baseline for the CSP task on the Alex-MP-20 dataset.

***De novo* generation.** For DNG models, we show the validity, coverage, and property metrics of the models in Tab. 2 and the stability, uniqueness, and novelty results in Tab. 3. Further ablation results across OMatG variants can be found in Tab. 10 in the Appendix, and qualitative plots of various distributions are compared in Appendix D.5. OMatG achieves state-of-the-art performance over DiffCSP, FlowMM, and MatterGen-MP for multiple positional interpolants thanks to the broader design space brought by the SIs. Figure 3 compares the distributions of the average energies above the hull for generated structures, exhibiting OMatG's superior performance for the generation of stable structures. OMatG consistently produces lower energy structures compared to previous models, and they are also generated close to their relaxed configuration. This, together with high novelty rates, begets improved S.U.N. rates. OMatG also outperforms FlowMM in settings where large language models are used as base distributions (Sriram et al., 2024), with results shown in Appendix E.

## 5.4. Discussion

**Crystal structure prediction.** We generally observe during hyperparameter optimization that the main learning challenge lies in the accurate prediction of the atomic coordinates, which tended to have a higher relative weight in calculating the full loss function in Eq. (32) (see Tabs 7 and 9 in the Appendix). However, the two best-performing models for the *perov-5* dataset instead exhibited the opposite—lending the most weight to learning of the cell vectors.

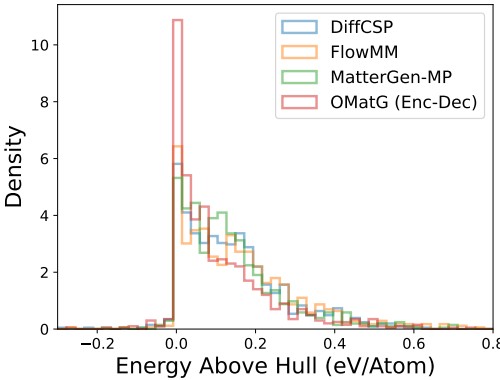

*Figure 3.* Histogram of the computed energies above the convex hull for structures generated by FlowMM, DiffCSP, MatterGen-MP, and OMatG (Enc-Dec interpolant). The OMatG model consistently produces lower energy structures compared to competing models. See Appendix D.4 for calculation details.

Our CSP results indicate a tradeoff between match rate and RMSE—which is only computed if matched. We find that as the number of generated structures matching known compositions in the test dataset increases, the structural fidelity—quantified by the accompanying RMSE—also tends to increase. This tradeoff most strongly influences our results on the *perov-5* dataset where all positional interpolants but the linear one can beat the previous state-of-the-art match rate. Here, particles generally find the correct local chemical configurations to flow towards during generation, but are not able to end up in the precise symmetric sites. The ODE-based linear interpolant without a latent variable, in contrast, has the lowest RMSE because the particles flow to more symmetric positions, but the local environments are not correct due to species mismatch. We quantify this effect in Fig. 7 in the Appendix.

For the CSP task on the *perov-5* dataset, we highlighted the particularly strong performance of the VP SBD and trigonometric interpolants in achieving a high match rate. Unlike other datasets, *perov-5* has a fixed number of $N = 5$ atoms per unit cell and a fixed (cubic) cell with varying side lengths and similar fractional positions—a combination which should not expose the model to a large variety of unit cell choices during interpolation or generation. By contrast, in other datasets, *no singular representation of the periodic repeat unit is imposed on flows*, meaning the model cannot learn the invariance (or even equivariance) to the choice of periodic repeat unit.[5] This likely contributes to the difficulty of unconstrained flow-based models in generating highly symmetric structures. Thus, the *perov-5* dataset presents a unique case where the invariance to unit cell choice does

---

[5]Using Niggli reduction during learning to enforce a unique choice of unit cell on structures from our datasets is not sufficient for enforcing this invariance during generation of structures.

not need to be learned, making this dataset a useful benchmark for evaluating positional interpolant performance. It is possible that the superior performance of the VP SBD and trigonometric interpolants arise from their ability to generate more circuitous flow trajectories compared to the strictly geodesic paths imposed by the linear interpolant—akin to the reasoning behind using latent variables to enhance learning in SIs (Albergo et al., 2023).

***De novo* generation.** Thorough hyperparameter optimization enables us to note trends among the best-performing DNG models. In Tabs 10 and 11 in the Appendix, we show the performance metrics and hyperparameters for each model by choice of positional interpolant, sampling scheme, and $\gamma(t)$ in the latent variable. We observe that several of our best performing models (with respect to S.U.N. and RMSD) possess lower levels of 'species noise' $\eta$ which sets the probability that an atom will change its identity if already in an unmasked state (see Appendix B.6). Additionally, we find that linear and trigonometric interpolants favor an element-order permutation as a data-dependent coupling during training (see Appendix B.8), while the encoder-decoder and SBD interpolants prefer to not use this coupling. Finally, we note that VP SBD models require a similar magnitude of velocity annealing during generation for positions and lattices (see Appendix B.7). This is in stark contrast to all other models, where a significantly larger velocity annealing parameter is required for generating the positions.

## 6. Conclusion

We adapt stochastic interpolants (SIs) for material generation tasks and propose Open Materials Generation (OMatG), a material-generation framework that unifies score-based diffusion and conditional flow-matching approaches under the umbrella of SIs. By incorporating an equivariant graph representation of crystal structures and explicitly handling periodic boundary conditions, OMatG jointly models spatial coordinates, lattice vectors, and discrete atomic species in a cohesive flow-based pipeline. Our extensive experiments on crystal structure prediction and *de novo* generation tasks demonstrate that OMatG sets a new state of the art in generative modeling for inorganic materials discovery, yielding more stable, novel, and unique structures than either pure diffusion or pure conditional flow-matching counterparts. We underscore the importance of flexible ML frameworks like OMatG, which can adapt to different types of materials datasets by optimizing the generative model accordingly. Our work represents a key step forward in applications of machine-learning methods to materials discovery. Looking forward, we plan to extend the flexibility of OMatG to additional interpolating functions, improve on the evaluation metrics and datasets, and investigate how different SIs influence the discovery of suitable materials.

# Acknowledgements

The authors thank Shenglong Wang at NYU IT High Performance Computing and Gregory Wolfe for their resourcefulness and valuable support. The authors acknowledge funding from NSF Grant OAC-2311632. P. H. and S. M. also acknowledge support from the Simons Center for Computational Physical Chemistry (Simons Foundation grant 839534, MT). The authors gratefully acknowledge use of the research computing resources of the Empire AI Consortium, Inc, with support from the State of New York, the Simons Foundation, and the Secunda Family Foundation. Moreover, the authors gratefully acknowledge the additional computational resources and consultation support that have contributed to the research results reported in this publication, provided by: IT High Performance Computing at New York University; the Minnesota Supercomputing Institute (http://www.msi.umn.edu) at the University of Minnesota; UFIT Research Computing (http://www.rc.ufl.edu) and the NVIDIA AI Technology Center at the University of Florida in part through the AI and Complex Computational Research Award; Drexel University through NSF Grant OAC-2320600.

# Impact Statement

This paper presents work whose goal is to advance the field of Machine Learning. There are many potential societal consequences of our work, none which we feel must be specifically highlighted here.

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

# A. Acronyms

Table 4 provides a list of acronyms used throughout this paper for reference.

*Table 4.* Acronym definitions.

| Acronym | Full Name |
|---------|-----------|
| AIRSS | Ab initio Random Structure Searching |
| CDVAE | Crystal Diffusion Variational Autoencoder |
| CFP | Chemical Formula Prediction |
| CFM | Conditional Flow Matching |
| CTMC | Continuous-Time Markov Chain |
| CSP | Crystal Structure Prediction |
| DFT | Density Functional Theory |
| DFM | Discrete Flow Matching |
| DNG | De Novo Generation |
| EGNN | Equivariant Graph Neural Network |
| GNoME | Graph Networks for Materials Exploration |
| MLIP | Machine-Learned Interatomic Potential |
| NequIP | Neural Equivariant Interatomic Potentials |
| ODE | Ordinary Differential Equation |
| OMatG | Open Materials Generation |
| RFM | Riemannian Flow Matching |
| SBD | Score-Based Diffusion |
| SBDM | Score-Based Diffusion Model |
| SDE | Stochastic Differential Equation |
| SI | Stochastic Interpolant |
| SMACT | Semiconducting Materials from Analogy and Chemical Theory |
| SUN | Stable, Unique, and Novel |

# B. Implementation Details of Stochastic Interpolants

### B.1. OMatG Framework

Figures 4 and 5 summarize the training and the integration pipeline of the OMatG framework, respectively. Depending on the specific task, there are several stochastic interpolants at once. For CSP, one stochastic interpolant considers lattice vectors $L$, and another one considers fractional coordinates $X$. The model output of CSPNet (see Appendix C.1) depends on the full structural representation $\{A, X, L\}$ and time $t$, where $A$ are the atomic species. For the DNG task, we additionally use discrete flow matching for the atomic species $A$ (Campbell et al., 2024).

During the numerical integration in the CSP task, $X$ and $L$ are integrated jointly while $A$ is fixed. For DNG, $A$ is evolved according to discrete flow matching (Campbell et al., 2024) (see Appendix B.6). For the SDE sampling scheme in Fig. 5, one chooses a time-dependent noise $\varepsilon(t)$ that only appears during integration and not during training (see Appendix B.4). Also, $\gamma(t)$ has to be unequal zero in order to prevent the divergence in $1/\gamma(t)$. However, since $\gamma(t)$ necessarily vanishes at times $t = 0$ and $t = 1$ (see Appendix B.2), one should choose a time-varying $\varepsilon(t)$ that vanishes near these endpoints (see Appendix B.4) (Albergo et al., 2023).

### B.2. Interpolant Choice

In this work, we are concerned with spatially linear interpolants of the form specified in Eq. (1). The following conditions must be met (Albergo et al., 2023):

$$\alpha(0) = \beta(1) = 1, \quad \alpha(1) = \beta(0) = \gamma(0) = \gamma(1) = 0, \quad \gamma(t) > 0 \ \forall t \in (0, 1). \tag{5}$$

Under these constraints, the form of the SI is relatively flexible, and many different interpolants can be defined. Also, the base distribution can be arbitrary as in CFM (Liu, 2022; Albergo & Vanden-Eijnden, 2023; Tong et al., 2024). In this work,

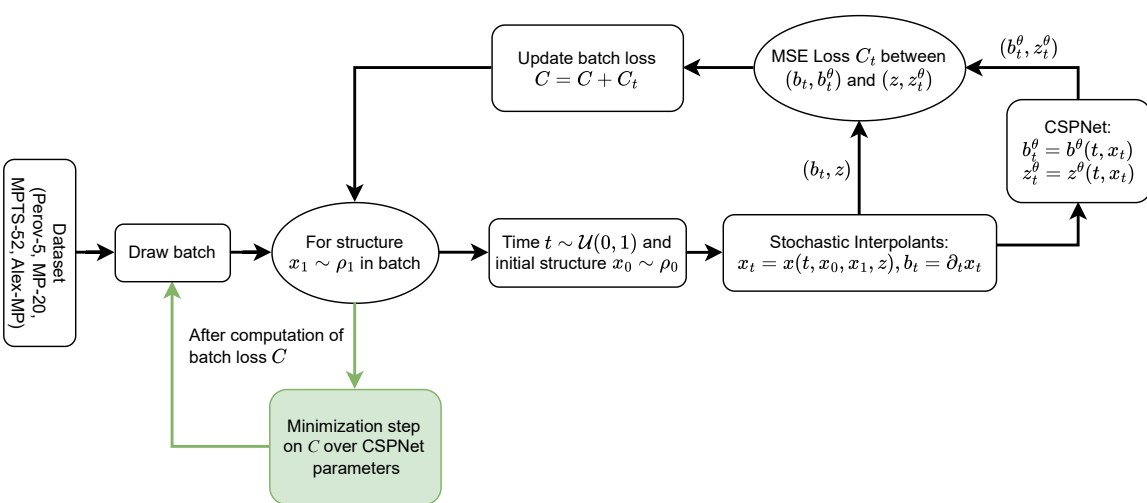

*Figure 4.* Training pipeline of the OMatG framework: A batch of structures is drawn from a dataset with target distribution $\rho_1$. Every structure $x_1 \sim \rho_1$ is connected with a structure $x_0$ from the base distribution $\rho_0$ with stochastic interpolants that yield the interpolated structure $x_t = x(t, x_0, x_1, z)$ and the drift $b_t = \partial_t x_t$ at time $t \sim \mathcal{U}(0, 1)$, possibly using a random variable $z \sim \mathcal{N}(0, \boldsymbol{I})$. The model CSPNet predicts $b_t^\theta = b^\theta(t, x_t)$ and $z_t^\theta = z(t, x_t)$ and its parameters are minimized based on the MSE losses in Eqs (2) and (3) [see also Eq. (32)].

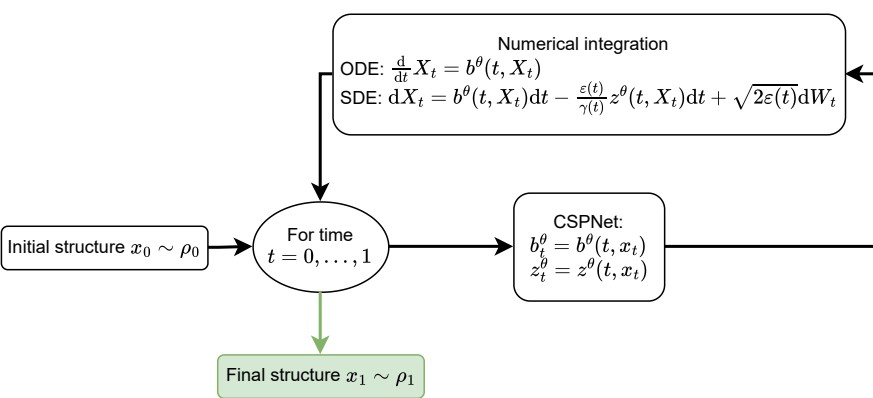

*Figure 5.* Numerical integration pipeline of the OMatG framework: An initial structure $x_0$ from the base distribution $\rho_0$ is numerically integrated following either an ODE or an SDE based on the model predictions $b_t^\theta$ and $z_t^\theta$. For an SDE, one can choose a noise $\varepsilon(t)$ during integration.

*Table 5.* SI parameters from Albergo et al. (2023).

| | **Stochastic Interpolant** | $\alpha(t)$ | $\beta(t)$ | $\gamma(t)$ |
|---|---|---|---|---|
| | linear | $1-t$ | $t$ | $\sqrt{at(1-t)}$ |
| Arbitrary $\rho_0$ | trig | $\cos\left(\frac{\pi}{2}t\right)$ | $\sin\left(\frac{\pi}{2}t\right)$ | $\sqrt{at(1-t)}$ |
| | enc-dec | $\cos^2(\pi t)1_{[0,\frac{1}{2})}(t)$ | $\cos^2(\pi t)1_{(\frac{1}{2},1]}$ | $\sin^2(\pi t)$ |
| Gaussian $\rho_0$ | VP SBD | $\sqrt{1-t^2}$ | $t$ | $0$ |

we mostly rely on interpolants originally defined in Albergo et al. (2023) and listed in Tab. 5. For the VP SBD interpolant, we allow for different widths $\sigma_0$ of the Gaussian base distribution $\rho_0$.

In Appendix B.9, we introduce additional VP and VE SBD interpolants beyond the one in Tab. 5 that are implemented in OMatG. The encoder-decoder (enc-dec) interpolant as defined in Tab. 5 evolves samples from the base distribution $\rho_0$ to follow an intermediate Gaussian distribution with variance 1 at the switch time $T_{\text{switch}} = 0.5$, before mapping them to a sample from $\rho_1$. This can be generalized to arbitrary variances $a > 0$ and switch times $T_{\text{switch}} \in (0, 1)$:

$$\alpha(t) = \cos^2\left(\frac{\pi(t - T_{\text{switch}}t)^p}{(T_{\text{switch}} - T_{\text{switch}}t)^p + (t - T_{\text{switch}}t)^p}\right) 1_{[0,T_{\text{switch}})}(t),$$

$$\beta(t) = \cos^2\left(\frac{\pi(t - T_{\text{switch}}t)^p}{(T_{\text{switch}} - T_{\text{switch}}t)^p + (t - T_{\text{switch}}t)^p}\right) 1_{(T_{\text{switch}},1]}(t), \tag{6}$$

$$\gamma(t) = \sqrt{a}\sin^2\left(\frac{\pi(t - T_{\text{switch}}t)^p}{(T_{\text{switch}} - T_{\text{switch}}t)^p + (t - T_{\text{switch}}t)^p}\right),$$

where $p \geq 1/2$. We consider the cases $p \in \{1/2, 1\}$ and note that the general interpolant in Eq. (6) reduces to the interpolant in Tab. 5 for $a = 1$, $p = 1$, and $T_{\text{switch}} = 0.5$.

### B.3. Antithetic Sampling

As shown by Albergo et al. (2023), the loss function can become unstable around $t = 0$ and $t = 1$ for certain choices of $\gamma(t)$. To account for this, we implement antithetic sampling. This requires simultaneously computing the loss at both $x^+$ and $x^-$ where

$$x^+(t, x_0, x_1, z) = \alpha(t)x_0 + \beta(t)x_1 + \gamma(t)z, \tag{7}$$

$$x^-(t, x_0, x_1, z) = \alpha(t)x_0 + \beta(t)x_1 - \gamma(t)z. \tag{8}$$

Both losses are computed using the same value of $z$ and subsequently averaged.

### B.4. Diffusion Coefficient

An important inference-time parameter for models integrated with an SDE is the choice of $\epsilon(t) \geq 0$ which plays the role of a diffusion coefficient. Albergo et al. (2023) note that the presence of $\gamma^{-1}(t)$ in the drift term seen in Fig. 5 can pose a numerical instability at the endpoints $t = 0$ and $t = 1$ during integration. For the choice $\epsilon_{\text{const}}(t) = c$, they consider integrating from some nonzero time $t \gtrsim 0$ to $t \lesssim 1$ in order to avoid the singularity. Alternatively, one can design a form for $\epsilon(t)$ such that it vanishes at these endpoints. In OMatG, we opt for the latter approach and consider a diffusion coefficient, $\epsilon_{\text{vanish}}(t)$, which vanishes at the endpoints

$$\epsilon_{\text{vanish}}(t) = \frac{c}{\left(1 + e^{-\frac{t-\mu}{\sigma}}\right)\left(1 + e^{-\frac{1-\mu-t}{\sigma}}\right)}. \tag{9}$$

Here, $c$ dictates the magnitude of the diffusion, $\mu$ sets the times at which the midpoints between $\epsilon(t) = 0$ and $\epsilon(t) = c$ are reached, and $\sigma$ controls the rate of this increase from $\epsilon(t) = 0$ to $\epsilon(t) = c$. The only constraints on these parameters are that $c \geq 0$, $\mu > 0$, and $\sigma > 0$. Importantly, these parameters should be chosen such that they are near zero at the endpoints.

## B.5. Interpolation with Periodic Boundary Conditions

We adopt a task-specific formulation for handling periodic boundary conditions with SIs tailored to flat tori, which are the relevant manifolds for fractional coordinates in crystal generation. We do not attempt to generalize stochastic interpolants (SIs) to arbitrary manifolds as in Riemannian flow matching (Chen & Lipman, 2024).

As in FlowMM (Miller et al., 2024), in order to uniquely define the interpolating paths, we rely on shortest geodesic interpolation paths between pairs of fractional coordinates from $x_0$ and $x_1$, ensuring that interpolants are well-defined and differentiable. As noted in Section 3.2.1, this shortest geodesic path is computed by first *unwrapping* one of the coordinates (say $x_1$) into its periodic image $x_1'$, such that it is the closest image to $x_0$. We then compute, for example, the linear interpolant $x(t, x_0, x_1') = (1 - t)x_0 + tx_1'$, as if in Euclidean space and finally wrap the interpolated path back onto the torus. (The geodesic is the same as the linear interpolant wrapped back into the box.)

We perform this procedure for all choices of interpolants. The reason for unwrapping according to the closest image first—for all interpolants—is because there are multiple ways to connect two points on a torus (*e.g.*, in a periodic box one can connect two points with or without crossing the box boundaries). All periodic stochastic interpolants are then defined this way, by computing $x(t, x_0, x_1', z) = \alpha(t)x_0 + \beta(t)x_1' + \gamma(t)z$ in the unwrapped (Euclidean) space and wrapping back onto the torus. We emphasize that this procedure is important not only for the choice of interpolant, but also for the addition of the latent variable $\gamma(t)z$ which also moves the interpolation trajectory away from the geodesic. Our process yields exactly the same shortest-path geodesic as in FlowMM if using the linear interpolant, and thus recovers its corresponding conditional flow-matching loss. We depict our implementation of periodic stochastic interpolants in Fig. 6. We also demonstrate in Fig. 6c that averaging over the latent variable $\gamma(t)z$ recovers the deterministic base interpolant path, as required by the SI framework.

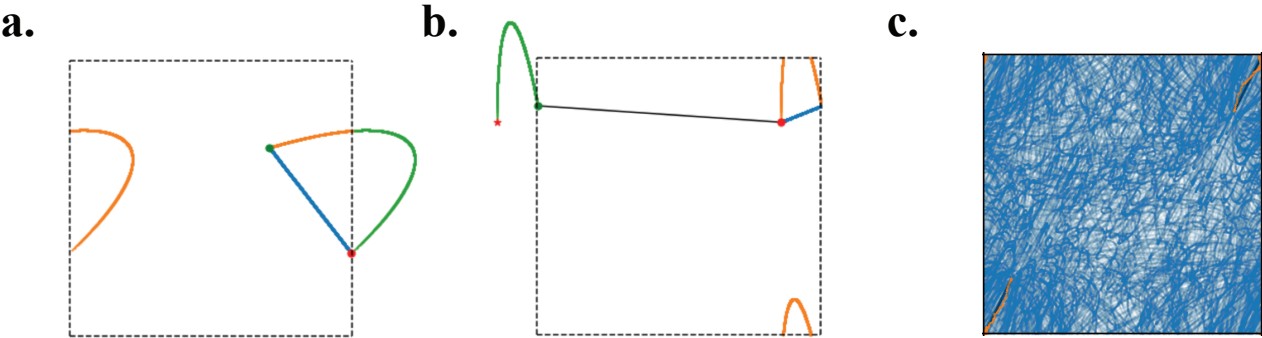

*Figure 6.* Extending interpolants to incorporate periodic boundary conditions. **(a–b)** The path for a score based diffusion interpolant is calculated by first computing the shortest-path geodesic (blue) between the initial (green dot) and final positions (red dot). Next, the path of the interpolant moving the final position outside the bounding box is computed (green), and finally the path is wrapped back into the bounding box to produce the interpolant trajectory (orange). **(c)** The effect of adding a latent variable to any interpolant must be handled similarly to calculating the path of a non-linear interpolant. For a linear interpolant with a nonzero $\gamma$, we show samples of possible paths (blue) and their averaged path (orange) which collapses onto the path of the linear interpolant.

## B.6. DFM Details

DFM allows for generative modeling of discrete sequences of tokens while respecting the discrete nature of the design space. As discussed, a parameterized neural network $p_{1|t}^{\theta}(x_1|x_t)$ is learned, which attempts to predict the final sequence from the sequence at time $t$. Borrowing from Campbell et al. (2024), we choose a conditional rate matrix $R_t(x_t, i|x_1)$—giving the rate of $x_t$ jumping to a different state $i$ given $x_1$—which generates the conditional flow $p_{t|1}(x_t|x_1)$ of the form:

$$R_t(x_t, i|x_1) = \frac{\text{ReLU}\left(\partial_t p_{t|1}(i|x_1) - \partial_t p_{t|1}(x_t|x_1)\right)}{S \cdot p_{t|1}(x_t|x_1)}, \tag{10}$$

where $S$ is the number of possible tokens a sequence element can take on. This conditional rate matrix can be modified by including a term that introduces stochasticity in the form of a detailed balance rate matrix $R_t^{DB}$ by writing $R_t^{\eta} = R_t + \eta R_t^{DB}$.

Here, (Campbell et al., 2024):

$$R_t^{DB}(i,j|x_1) = \eta\delta\{i,x_1\}\delta\{j,M\} + \frac{\eta t}{1-t}\delta\{i,M\}\delta\{j,x_1\}, \tag{11}$$

where $M$ is the masking token. The parameter $\eta \in \mathbb{R}^+$ represents the level of stochasticity that only appears during generation.

During generation, our objective is to compute $R_t^\theta(x_t, i)$ based on the learned distribution $p_{1|t}^\theta(x_1|x_t)$. Formally, we have

$$R_t^\theta(x_t, i) = \mathbb{E}_{p_{1|t}^\theta(x_1|x_t)}\left[R_t^\eta(x_t, i|x_1)\right] \tag{12}$$

In practice, Campbell et al. (2024) show that we need not compute a full expectation, but rather, simply draw $x_1 \sim p_{1|t}^\theta(x_1|x_t)$, evaluate the conditional rate matrix $R_t^\eta(x_t, i|x_1)$, and perform an update of $x_t$ to $x_{t+\Delta t}$ with discrete time step $\Delta t$ directly from this by sampling $x_{t+\Delta t}$ according to

$$p_{t+\Delta t|t}(x_{t+\Delta t}|x_1, x_t) = \delta\{x_t, x_{t+\Delta t}\} + R_t^\eta(x_t, i|x_1)\Delta t. \tag{13}$$

### B.7. Velocity Annealing

Velocity annealing—rescaling the learned velocity field during generation to increase velocity over time as $b^\theta(t,x) \to (1+st)\,b^\theta(t,x)$ with $s$ as an hyperparameter during integration—has been empirically shown to improve performance in a number of studies that apply CFM to physical systems (Yim et al., 2023; Bose et al., 2024; Miller et al., 2024). For instance, Miller et al. (2024) demonstrated that applying velocity annealing significantly improves performance in CSP and DNG benchmarks for materials. Motivated by these findings, we include velocity annealing in OMatG as a tunable hyperparameter, while emphasizing that this technique lacks a formal theoretical justification within the mathematical frameworks underlying flow models and stochastic interpolants.

### B.8. Data-Dependent Coupling

SIs have been used with data-dependent couplings (Albergo et al., 2024), where a coupling function $\nu(x_0, x_1)$ enables biasing of $x_0$ based on the sampled $x_1$. In OMatG, we incorporate an optional data-dependent coupling that enforces an ordering (*i.e.*, a permutation on the order of atomic elements within a structure) that produces the minimum fractional-coordinate distance between each particle pair $(\boldsymbol{x}_0^i, \boldsymbol{x}_1^i)$ from structures $x_0 \in \rho_0$ and $x_1 \in \rho_1$. We find that the inclusion of this data-dependent coupling is optimal during hyperparameter tuning depending on the type of model: CSP models typically performed better without this coupling, but DNG models (see Tab. 11) can benefit in certain cases from minimizing traveled distance *via* permutation of elements.

Formally, our coupling is conditional on the sampled $(x_0, x_1)$ and is defined as

$$\arg\min_p \sum_i d(p(x_0^i), x_1^i). \tag{14}$$

Here, $d(\cdot, \cdot)$ is a distance metric which we define on a periodic manifold in fractional-coordinate space (*i.e.*, a four-dimensional torus) and $p$ is some permutation function that permutes the discrete indices $i$. Under this coupling, we still sample $(x_0, x_1)$ independently but then bias the sampled $x_0$ to travel the minimum permutational distance necessary to reach the target structure.

### B.9. SI unifies CFM and SBDM

The SI framework implemented in OMatG unifies the frameworks of CFM, as implemented in FlowMM (Miller et al., 2024), and SBDM, as implemented in DiffCSP (Jiao et al., 2023) and MatterGen (Zeni et al., 2025). FlowMM is naturally subsumed by OMatG. For the choice of ODE-based sampling, the velocity term $b^\theta(t,x)$ is learned by minimizing the loss function in Eq. (2). By using $\gamma(t) = 0$ in the linear interpolant $x(t, x_0, x_1) = (1-t)x_0 + tx_1$ (see Appendix B.2), Eq. (2) becomes identical to the FlowMM loss (see Eq. (15) in Miller et al. (2024)). Furthermore, the treatment of periodic boundary conditions for the linear interpolant (see Appendix B.5) leads to the same geodesic paths as in FlowMM, and the center-of-mass motion of the ground-truth velocity is removed similarly in both frameworks.

The connection between SIs and SBDM requires the discussion of both variance-preserving (VP) and variance-exploding (VE) cases. In the VP case (Sohl-Dickstein et al., 2015; Ho et al., 2020), a sequence of $N$ noise increments with variance $\beta_i$ perturbs data $y_0$ as

$$y_i = \sqrt{1 - \beta_i}\, y_{i-1} + \sqrt{\beta_i}\, z_{i-1}, \quad i = 1, \ldots, N, \tag{15}$$

where $z_i \sim \mathcal{N}(0, \boldsymbol{I})$. As $N \to \infty$, this converges to the SDE

$$\mathrm{d}y_s = -\frac{1}{2}\beta(s)y_s\,\mathrm{d}t + \sqrt{\beta(s)}\,\mathrm{d}w_s, \tag{16}$$

where $w_s$ is the standard Wiener process (Song et al., 2021). Under this stochastic process, the data distribution at time $s = 0$ is transformed into a Gaussian base distribution as $s \to \infty$. This differs from the time convention in SI where samples from the base distribution at time $t = 0$ are transported to samples from the data distribution at $t = 1$. With a corresponding change of variables $s(t) = -\log(t)$, Aranguri et al. (2025) show within the SI framework that $y_s$ is equal in law to the one-sided interpolant

$$x(t, x_0, x_1) = \sqrt{1 - \tau^2(t)}\, x_0 + \tau(t)x_1, \tag{17}$$

where

$$\tau(t) = \exp\left(-\frac{1}{2}\int_0^{-\log(t)} \beta(u)\,\mathrm{d}u\right). \tag{18}$$

In accordance with SBDM, the base distribution $\rho_0$ is Gaussian, that is, $x_0 \sim \mathcal{N}(0, \sigma_0^2 \boldsymbol{I})$. Here, we introduced the width $\sigma_0$ of the base distribution as a tunable hyperparameter.

The SDE in Eq. (16) and the variance-schedule $\beta(s)$ are often considered on the time interval $s \in [0, 1]$, which implies that $\tau(t)$ from Eq. (18) is only used in $t \in [1/e, 1]$ (Aranguri et al., 2025). OMatG implements three such schedules. The linear schedule (Ho et al., 2020; Aranguri et al., 2025) is $\beta^{\mathrm{lin}}(s) = \beta_{\min} + s(\beta_{\max} - \beta_{\min})$ where $\beta_{\min}$ and $\beta_{\max}$ are chosen empirically. This implies

$$\tau^{\mathrm{lin}}(t) = \exp\left[\frac{1}{2}\beta_{\min}\log(t) - \frac{1}{4}(\beta_{\max} - \beta_{\min})\log^2(t)\right]. \tag{19}$$

This function is well-behaved for the entire time range $t \in [0, 1]$ and implemented as such in OMatG.

For the cosine schedule, which is used in DiffCSP for the lattice vectors, the noise variance in Eq. (15) is given by $\beta_i = 1 - \bar{\alpha}_i/\bar{\alpha}_{i-1}$ with

$$\bar{\alpha}_i = \frac{f(i)}{f(0)}, \quad f(i) = \cos^2\left(\frac{\pi}{2}\frac{i/N + d}{1 + d}\right), \tag{20}$$

where $d$ is a small constant offset (Nichol & Dhariwal, 2021). As $N \to \infty$, one gets for $s \in [0, 1]$

$$\beta^{\cos}(s) = -\frac{\mathrm{d}}{\mathrm{d}s}\log\left[\frac{\cos^2\left(\frac{\pi}{2}\frac{s+d}{1+d}\right)}{\cos^2\left(\frac{\pi}{2}\frac{d}{1+d}\right)}\right] = \frac{\pi}{1+d}\tan\left(\frac{\pi}{2}\frac{s+d}{1+d}\right). \tag{21}$$

For $t \in [1/e, 1]$, this leads to

$$\tau^{\cos}(t) = \csc\left[\frac{\pi}{2+2d}\right]\sin\left[\frac{\pi + \pi\log(t)}{2+2d}\right], \tag{22}$$

and for $t \in [0, 1/e)$, we use $\tau^{\cos}(t) = 0$.

The schedule $\tau^{\mathrm{const}}(t) = t$ on $t \in [0, 1]$ corresponds to a constant schedule $\beta^{\mathrm{const}}(s) = 2$ and yields the SBD interpolant of Appendix B.2 derived in Albergo et al. (2023). This is the VP SBD interpolant considered throughout this paper.

In the VE case (Song & Ermon, 2019), a sequence of $N$ noise increments with variance $\sigma_i$ perturbs data $y_0$ as

$$y_i = y_{i-1} + \sqrt{\sigma_i^2 - \sigma_{i-1}^2}\, z_{i-1}. \tag{23}$$

As $N \to \infty$, this converges to the SDE (Song et al., 2021)

$$\mathrm{d}y_s = \sqrt{\frac{\mathrm{d}[\sigma^2(s)]}{\mathrm{d}s}}\,\mathrm{d}w_s. \tag{24}$$

The corresponding one-sided interpolant in the SI framework is given by

$$x_t = \sqrt{\sigma^2(1-t) - \sigma^2(0)}\, x_0 + x_1, \tag{25}$$

where again $x_0 \sim \mathcal{N}(0, \sigma_0^2 \boldsymbol{I})$ (Aranguri et al., 2025). As in DiffCSP for the fractional coordinates, the schedule $\sigma(s)$ on $s \in [0,1]$ is typically given by $\sigma(s) = \sigma_{\min}(\sigma_{\max}/\sigma_{\min})^s$ which we implement in OMatG. The parameters $\sigma_{\min}$ and $\sigma_{\max}$ are optimizeable hyperparameters. The reported match rate and RMSE of the CSP task for the VE SBD positional interpolant for the *MP-20* dataset in Tab. 1 are close to the ones of DiffCSP. This highlights that OMatG is able to reproduce similar conditions to those in DiffCSP.

### B.10. Comparison of OMatG-Linear to FlowMM

A subset of OMatG models, specifically those which use linear interpolants for both the fractional coordinates and lattice vectors, map closely onto the conditional flow-matching model FlowMM (Miller et al., 2024). The notable differences between OMatG-Linear models and FlowMM are as follows: (1) Discrete flow matching on species for OMatG *vs.* analog bits for FlowMM. (2) Lattice matrix representation for OMatG *vs.* lattice parameter representation (lengths and angles) for FlowMM. (3) Original CSPNet encoder for OMatG *vs.* slightly modified CSPNet for FlowMM.

OMatG's CSP results improve upon FlowMM's. Since the CSP task does not utilize any species learning, the different handling of species is not sufficient to fully explain the differences in model performance for CSP. For DNG models, the handling of species is also a relevant difference between OMatG-Linear and FlowMM.

## C. Model Architecture

### C.1. Graph Neural Network

We implement a message-passing graph neural network (GNN) with CSPNet as introduced in Jiao et al. (2023):

$$\boldsymbol{h}_{(0)}^i = \phi_{\boldsymbol{h}_{(0)}}(\boldsymbol{a}^i) \tag{26}$$

$$\boldsymbol{m}_{(s)}^{ij} = \varphi_m\left(\boldsymbol{h}_{s-1}^i, \boldsymbol{h}_{s-1}^j, \boldsymbol{l}, \text{SinusoidalEmbedding}(\boldsymbol{x}^j - \boldsymbol{x}^i)\right) \tag{27}$$

$$\boldsymbol{m}_{(s)}^i = \sum_{j=1}^N \boldsymbol{m}_{(s)}^{ij} \tag{28}$$

$$\boldsymbol{h}_{(s)}^i = \boldsymbol{h}_{(s-1)}^i + \varphi_h(\boldsymbol{h}_{(s-1)}, \boldsymbol{m}_{(s)}^i) \tag{29}$$

$$b_{\boldsymbol{x}} = \varphi_{\boldsymbol{x}}\left(\boldsymbol{h}_{(\max s)}^i\right) \tag{30}$$

$$b_{\boldsymbol{l}} = \varphi_{\boldsymbol{l}}\left(\frac{1}{n}\sum_{i=1}^n \boldsymbol{h}_{(\max s)}^i\right) \tag{31}$$

Here, node embeddings $\boldsymbol{h}_{(s)}^j$ of node $j$ at layer $s$ are initialized as a function of the atom types, $\boldsymbol{a}$. Embeddings are then updated by a message passing scheme through a series of graph convolution layers. Messages are computed with a parameterized neural network, $\varphi_m$, from neighboring node embeddings as well as information about the lattice, $\boldsymbol{l}$, and distance between the fractional coordinates $\boldsymbol{x}$. All necessary drift and denoiser terms are computed from single layer MLPs applied to the final node embeddings.

For the CFP model that should only predict compositions, we simply remove the input of the lattice $\boldsymbol{l}$ and the fractional coordinates $\boldsymbol{x}$ from the computation of the message in Eq. (27). This ensures that the output $p_{1|t}^\theta(\boldsymbol{a}_1|x_t)$ of CSPNet for the composition does not depend on lattice vectors or fractional coordinates, while preserving permutational equivariance.

## C.2. Loss Function

With Eqs (2), (3), and (4), we can construct a loss function for the modeling of our joint distribution of interest for the DNG task,

$$
\begin{aligned}
\mathcal{L}(\theta) =\mathbb{E}_{t,z,x_0,x_1}\Big[ & \\
& \lambda_{\boldsymbol{x},b}\left[|b_{\boldsymbol{x}}^{\theta}(t,x_t)|^2 - 2\partial_t x(t,x_0,x_1,z)\cdot b_{\boldsymbol{x}}^{\theta}(t,x_t)\right] + \lambda_{\boldsymbol{x},z}\left[|z_{\boldsymbol{x}}^{\theta}(t,x_t)|^2 - 2z_{\boldsymbol{x}}^{\theta}(t,x_t)\cdot z\right] \\
& + \lambda_{\boldsymbol{l},b}\left[|b_{\boldsymbol{l}}^{\theta}(t,x_t)|^2 - 2\partial_t x(t,x_0,x_1,z)\cdot b_{\boldsymbol{l}}^{\theta}(t,x_t)\right] + \lambda_{\boldsymbol{l},z}\left[|z_{\boldsymbol{l}}^{\theta}(t,x_t)|^2 - 2z_{\boldsymbol{l}}^{\theta}(t,x_t)\cdot z\right] \\
& + \lambda_{\boldsymbol{a}}\left[\log p_{1|t}^{\theta}(\boldsymbol{a}_1|x_t)\right]\Big].
\end{aligned}
\tag{32}
$$

For the CSP task, the last line is left out. The $\lambda$ terms correspond to the relative weights of each term in the loss function. These weighting factors are hyperparameters that are included in our hyperparameter sweep. The respective terms for the fractional coordinates and lattice vectors corresponding to Eqs (2) and (3) are equivalent to a mean-squared error (MSE) loss function as, for instance,

$$
\mathcal{L}_b^{\mathrm{MSE}}(\theta) = \mathbb{E}_{t,z,x_0,x_1}\left[|b^{\theta}(t,x_t) - \partial_t x(t,x_0,x_1,z)|^2\right]
\tag{33}
$$

for the velocity term. They only differ by a constant term that does not influence gradients. We do not include that constant term because the possible divergence of $\partial_t \gamma(t)$ near $t = 0$ and $t = 1$ can artificially inflate the absolute value of the loss, even when antithetic sampling is applied (see Section B.3).

## C.3. Hyperparameter Optimization

For every choice of the positional interpolant, sampling scheme, and latent variable $\gamma$, an independent hyperparameter optimization was performed using the `Ray Tune` package (Liaw et al., 2018) in conjunction with the HyperOpt Python library (Bergstra et al., 2013) for Bayesian optimization. The tuned hyperparameters include both those relevant during training—the relative loss weights $\tilde{\lambda}$, the choice of stochastic interpolant for the lattice vectors, the parameters for chosen $\gamma(t)$ (if necessary), the sampling scheme, the usage of data-dependent coupling, the batch size, and the learning rate—and during inference—the number of integration steps, the choice of the noises $\varepsilon(t)$ and $\eta$, and the magnitude of the velocity annealing parameter $s$ for both lattice vectors and atomic coordinates. Hyperparameters are sampled according to the distributions below[6]:

- Number of integration timesteps $\sim \mathrm{Uniform}(100, 1000)$.

- Batch size $\sim \mathrm{Choice}(32, 64, 128, 256, 512, 1024)$.

- Min. permutational distance data coupling $\sim \mathrm{Choice}(\mathrm{True}, \mathrm{False})$.

- Relative weigths $\tilde{\lambda}_{\mathbf{x},b}, \tilde{\lambda}_{\mathbf{x},z}, \tilde{\lambda}_{\mathbf{a}} \sim \mathrm{LogUniform}(0.1, 2000.0)$.

- Relative weight $\tilde{\lambda}_{\mathbf{l},z} \sim \mathrm{LogUniform}(0.1, 100.0)$.

- Niggli reduction of cell during training $\sim \mathrm{Choice}(\mathrm{True}, \mathrm{False})$.

- DFM Stochastictity $\sim \mathrm{Uniform}(0, 50.0)$.

- Learning rate $\sim \mathrm{LogUniform}(10^{-5}, 10^{-2})$.

- Weight decay $\sim \mathrm{LogUniform}(10^{-5}, 10^{-3})$.

- Velocity annealing coefficient (both for $\mathbf{x}$ and $\mathbf{l}$) $\sim \mathrm{Uniform}(0.0, 15.0)$.

- Diffusion coefficient parameter $c$ (both for $\mathbf{x}$ and $\mathbf{l}$) $\sim \mathrm{Uniform}(0.1, 10.0)$.

- Diffusion coefficient parameter $\mu$ (both for $\mathbf{x}$ and $\mathbf{l}$) $\sim \mathrm{Uniform}(0.05, 0.3)$.

- Diffusion coefficient parameter $\sigma$ (both for $\mathbf{x}$ and $\mathbf{l}$) $\sim \mathrm{Uniform}(0.005, 0.05)$.

---

[6]Relative loss weights for $\mathbf{a}$ are only swept over for DNG. Otherwise only weight parameters for $\mathbf{x}$ and $\mathbf{l}$ are optimized. Relative loss weights for the denoiser are only included when SDE integration is used.

- Parameter $T_{\text{switch}}$ of encoder-decoder interpolant $\sim \text{Uniform}(0.1, 0.9)$.

- Parameter $p$ of encoder-decoder interpolant $\sim \text{Choice}(0.5, 1.0)$.

- Parameter $a$ of $\gamma(t)$ functions $\sim \text{LogUniform}(0.01, 10.0)$.

- Standard deviation $\sigma_0$ of Gaussian $\rho_0$ for SBD interpolants $\sim \text{LogUniform}(0.01, 10.0)$.

- Parameter $\sigma_{\min}$ of the $\sigma(s)$ schedule of the VE SBD interpolant $\sim \text{Uniform}(0.001, 0.01)$.

- Parameter $\sigma_{\max}$ of the $\sigma(s)$ schedule of the VE SBD interpolant $\sim \text{Uniform}(0.1, 1.0)$.

The relative loss weight for the velocity of the lattice vectors is fixed: $\tilde{\lambda}_{1,b} = 1$. By ensuring that the sum of all relevant loss weights $\lambda$ is one, one can transform the relative weights $\tilde{\lambda}$ to the weights $\lambda$ in Eq. (32).

For the CSP models, the hyperparameter optimization attempts to maximize the match rate. For the DNG models, we combine the metrics in Tab. 2 to a single evaluation metric $\text{eval}_{\text{DNG}}$ that is supposed to be minimized by the hyperparameter optimization:

$$
\text{eval}_{\text{DNG}} = \text{avg}\Bigg[ \text{combined validity},
$$
$$
\text{avg}\Big[ \text{wdist}(\rho), \text{wdist}(N\text{ary}), \text{wdist}(\langle CN \rangle) \Big],
$$
$$
\text{avg}\Big[ 1 - \text{coverage recall}, 1 - \text{coverage precision} \Big] \Bigg]. \tag{34}
$$

Here, the function avg returns the average of its arguments.

We perform hyperparameter optimization for the DNG task only for the *MP-20* dataset. For the CSP task, we optimize hyperparameters for the *perov-5* and *MP-20* datasets. For the CSP task on the *MPTS-52* and *Alex-MP-20* datasets, we simply transfer the hyperparameters of the optimized *MP-20* models. We provide hyperparameter-tuned models with the relevant performance metrics and hyperparameters for *perov-5* CSP in Tabs 6 and 7, *MP-20* CSP in Tabs 8 and 9, and *MP-20* DNG in Tabs 10 and 11.

Many models were *partially* trained and compared in the process of hyperparameter tuning: on average 27 models (perov-5) and 32 models (MP-20) for each choice of positional interpolant, sample scheme, and latent variable.

*Table 6.* Study for the *perov-5* dataset comparing CSP performance metrics for choice of positional interpolant, sample scheme, and $\gamma(t)$ in the latent variable (or width $\sigma_0$ of the Gaussian base distribution $\rho_0$ for the SBD interpolants, and parameters $\sigma_{\min}$ and $\sigma_{\max}$ of the $\sigma(s)$ schedule of the VE SBD interpolant).

| Positional interpolant | Positional sampling scheme | Positional $\gamma(t)$ | Match rate (%, Full / Valid) | RMSE (Full / Valid) |
|---|---|---|---|---|
| Linear | ODE | None: $\gamma = 0$ | 51.86% / 50.62% | 0.0757 / 0.0760 |
| Linear | ODE | LatentSqrt: $\gamma = \sqrt{0.034\,t\,(1-t)}$ | 72.21% / 62.54% | 0.3510 / 0.3444 |
| Linear | SDE | LatentSqrt: $\gamma = \sqrt{0.028\,t\,(1-t)}$ | 74.16% / 72.87% | 0.3307 / 0.3315 |
| Trigonometric | ODE | None: $\gamma = 0$ | 81.51% / 52.36% | 0.3674 / 0.3628 |
| Trigonometric | ODE | LatentSqrt: $\gamma = \sqrt{0.011\,t\,(1-t)}$ | 80.85% / 79.55% | 0.3864 / 0.3873 |
| Trigonometric | SDE | LatentSqrt: $\gamma = \sqrt{0.063\,t\,(1-t)}$ | 73.37% / 71.60 % | 0.3610 / 0.3614 |
| Encoder-Decoder | ODE | Enc-Dec: $\gamma = \sqrt{0.66}\sin^2\left(\frac{\pi(t-0.80t)}{(0.80-0.80t)+(t-0.80t)}\right)$ | 68.08% / 64.60% | 0.4005 / 0.4003 |
| Encoder-Decoder | SDE | Enc-Dec: $\gamma = \sqrt{8.45}\sin^2\left(\frac{\pi(t-0.61t)}{(0.61-0.61t)+(t-0.61t)}\right)$ | 78.28% / 76.80% | 0.3616 / 0.3620 |
| VP Score-Based Diffusion | ODE | $\sigma_0 = 0.28$ | 83.06% / 81.27% | 0.3753 / 0.3755 |
| VP Score-Based Diffusion | SDE | $\sigma_0 = 0.13$ | 76.54% / 64.46% | 0.3529 / 0.3402 |
| VE Score-Based Diffusion | ODE | $\sigma_0 = 8.96$; $\sigma_{\min} = 0.0078$, $\sigma_{\max} = 0.5165$ | 60.18% / 52.97% | 0.2510 / 0.2337 |

*Table 7.* Study for the *perov-5* dataset CSP comparing hyperparameters for each choice of positional interpolant, sample scheme, and $\gamma(t)$ (as reported in Tab. 6).

| Pos. interpolant, Sampling scheme, $\gamma$ | Cell interpolant Sampling scheme, $\gamma(t)$ | Annealing param. $s$ (Pos. / Cell) | Integration steps | Min. dist. permutation | Niggli | $\lambda_{\boldsymbol{x},b}/\lambda_{\boldsymbol{l},b}/\lambda_{\boldsymbol{x},z}/\lambda_{\boldsymbol{l},z}$ |
|---|---|---|---|---|---|---|
| Linear, ODE, None | Linear, ODE, $\gamma = 0$ | 14.11 / 2.90 | 820 | False | False | 0.9729 / 0.0271 / - / - |
| Linear, ODE, LatentSqrt | Linear, ODE, $\gamma = 0$ | 0.008 / 12.19 | 820 | True | True | 0.9724 / 0.0276 / - / - |
| Linear, SDE, LatentSqrt | Linear, ODE, $\gamma = \sqrt{0.013\,t\,(1-t)}$ | 8.20 / 1.46 | 910 | True | True | 0.0024 / 0.0051 / 0.9925 / - |
| Trig, ODE, None | Linear, ODE, $\gamma = \sqrt{0.021\,t\,(1-t)}$ | 14.99 / 14.97 | 880 | True | False | 0.9983 / 0.0017 / - / - |
| Trig, ODE, LatentSqrt | Linear, ODE, $\gamma = 0$ | 9.68 / 2.42 | 110 | False | False | 0.1130 / 0.8870 / - / - |
| Trig, SDE, LatentSqrt | Linear, ODE, $\gamma = \sqrt{0.051\,t\,(1-t)}$ | 3.43 / 0.03 | 900 | True | True | 0.6868 / 0.0643 / 0.2489 / - |
| Enc-Dec, ODE, Enc-Dec | Linear, ODE, $\gamma = 0$ | 14.94 / 0.318 | 460 | True | True | 0.8563 / 0.1437 / - / - |
| Enc-Dec, SDE, Enc-Dec | Linear, ODE, $\gamma = \sqrt{0.154\,t\,(1-t)}$ | 14.55 / 0.075 | 930 | True | False | 0.2828 / 0.0004 / 0.7168 / - |
| VP SBD, ODE | SBD, SDE, $\sigma = 0.61$ | 12.79 / 2.69 | 130 | True | True | 0.0035 / 0.0121 / - / 0.9844 |
| VP SBD, SDE | Trig, SDE, $\gamma = \sqrt{0.029\,t\,(1-t)}$ | 11.54 / 11.53 | 350 | True | False | 0.2898 / 0.1960 / 0.3259 / 0.1883 |
| VE SBD, ODE | Trig, SDE, $\gamma = \sqrt{0.024\,t\,(1-t)}$ | 0.003 / 14.93 | 380 | False | False | 0.9800 / 0.0187 / - / 0.0014 |

*Table 8.* Study for the *MP-20* dataset comparing CSP performance metrics for choice of positional interpolant, sample scheme, and $\gamma(t)$ in the latent variable (or width $\sigma_0$ of the Gaussian base distribution $\rho_0$ for the SBD interpolants, and parameters $\sigma_{\min}$ and $\sigma_{\max}$ of the $\sigma(s)$ schedule of the VE SBD interpolant).

| Positional interpolant | Positional sampling scheme | Positional $\gamma(t)$ | Match rate (%, Full / Valid) | RMSE (Full / Valid) |
|---|---|---|---|---|
| Linear | ODE | None: $\gamma = 0$ | 69.83% / 63.75% | 0.0741 / 0.0720 |
| Linear | ODE | LatentSqrt: $\gamma = \sqrt{0.258\,t\,(1-t)}$ | 55.60% / 50.04% | 0.1531 / 0.1494 |
| Linear | SDE | LatentSqrt: $\gamma = \sqrt{0.063\,t\,(1-t)}$ | 68.20% / 61.88% | 0.1632 / 0.1611 |
| Trigonometric | ODE | None: $\gamma = 0$ | 65.30% / 58.94% | 0.1184 / 0.1149 |
| Trigonometric | ODE | LatentSqrt: $\gamma = \sqrt{0.033\,t\,(1-t)}$ | 66.19% / 59.81% | 0.1002 / 0.0968 |
| Trigonometric | SDE | LatentSqrt: $\gamma = \sqrt{0.049\,t\,(1-t)}$ | 68.90% / 62.65% | 0.1249 / 0.1235 |
| Encoder-Decoder | ODE | Enc-Dec: $\gamma = \sqrt{1.99}\sin^2\left(\frac{\pi(t-0.65t)}{(0.65-0.65t)+(t-0.65t)}\right)$ | 55.15% / 49.45% | 0.1306 / 0.1260 |
| Encoder-Decoder | SDE | Enc-Dec: $\gamma = \sqrt{0.04}\sin^2\left(\frac{\pi(t-0.42t)^{0.5}}{(0.42-0.42t)^{0.5}+(t-0.42t)^{0.5}}\right)$ | 57.69% / 52.44% | 0.1160 / 0.1125 |
| VP Score-Based Diffusion | ODE | $\sigma_0 = 0.22$ | 45.57% / 39.48% | 0.1880 / 0.1775 |
| VP Score-Based Diffusion | SDE | $\sigma_0 = 2.29$ | 42.29% / 38.08% | 0.2124 / 0.2088 |
| VE Score-Based Diffusion | ODE | $\sigma_0 = 9.77;\ \sigma_{\min} = 0.0047,\ \sigma_{\max} = 0.9967$ | 63.79% / 57.82% | 0.0809 / 0.0780 |

*Table 9.* Study for the *MP-20* dataset comparing CSP hyperparameters for choice of positional interpolant, sample scheme, and $\gamma(t)$ (as reported in Tab. 8).

| Pos. interpolant, Sampling scheme, $\gamma$ | Cell interpolant Sampling scheme, $\gamma(t)$ | Annealing param. $s$ (Pos. / Cell) | Integration steps | Min. dist. permutation | Niggli | $\lambda_{x,b}/\lambda_{l,b}/\lambda_{x,z}/\lambda_{l,z}$ |
|---|---|---|---|---|---|---|
| Linear, ODE, None | Linear, ODE, $\gamma = 0$ | 10.18 / 1.82 | 210 | False | False | 0.9994 / 0.0006 / - / - |
| Linear, ODE, LatentSqrt | Trig, ODE, $\gamma = \sqrt{2.976\,t\,(1-t)}$ | 7.76 / 4.12 | 690 | False | True | 0.9976 / 0.0024/ - / - |
| Linear, SDE, LatentSqrt | Linear, SDE, $\gamma = \sqrt{0.132\,t\,(1-t)}$ | 11.58 / 5.08 | 310 | False | False | 0.0073 / 0.0642 / 0.9154 / 0.0131 |
| Trig, ODE, None | Enc-Dec, SDE, $\gamma = \sqrt{5.27}\sin^2\left(\frac{\pi(t-0.41t)^{0.5}}{(0.41-0.41t)^{0.5}+(t-0.41t)^{0.5}}\right)$ | 12.34 / 3.61 | 170 | False | False | 0.9967 / 0.0023 / - / 0.0010 |
| Trig, ODE, LatentSqrt | Linear, SDE, $\gamma = \sqrt{0.017\,t\,(1-t)}$ | 13.54 / 2.38 | 780 | False | True | 0.9830 / 0.0167 / - / 0.0003 |
| Trig, SDE, LatentSqrt | Trig, ODE, $\gamma = 0$ | 11.48 / 0.43 | 740 | True | True | 0.2468 / 0.0301 / 0.7231 / - |
| Enc-Dec, ODE, Enc-Dec | Trig, SDE, $\gamma = \sqrt{0.219\,t\,(1-t)}$ | 12.29 / 4.30 | 820 | False | True | 0.6892 / 0.1235 / - / 0.1873 |
| Enc-Dec, SDE, Enc-Dec | Linear, ODE, $\gamma = \sqrt{4.961\,t\,(1-t)}$ | 3.78 / 1.14 | 710 | False | True | 0.6143 / 0.0063 / 0.3794 / - |
| VP SBD, ODE | Linear, ODE, $\gamma = 0$ | 6.61 / 2.45 | 890 | True | True | 0.9598 / 0.0402 / - / - |
| VP SBD, SDE | Linear, ODE, $\gamma = \sqrt{3.684\,t\,(1-t)}$ | 6.46 / 0.67 | 600 | True | True | 0.6060 / 0.0112 / 0.3828 / - |
| VE SBD, ODE | Linear, SDE, $\gamma = \sqrt{0.017\,t\,(1-t)}$ | 8.28 / 0.43 | 660 | False | False | 0.9813 / 0.0005 / - / 0.0182 |

*Table 10.* Study for the *MP-20* dataset comparing DNG performance metrics for choice of positional interpolant, sample scheme, and $\gamma(t)$ in the latent variable (or width $\sigma_0$ of the Gaussian base distribution $\rho_0$ for the SBD interpolants, and parameters $\sigma_{\min}$ and $\sigma_{\max}$ of the $\sigma(s)$ schedule of the VE SBD interpolant). S.U.N. rates are computed according to the MatterSim potential.

| Positional interpolant | Positional sampling scheme | Positional $\gamma(t)$ | S.U.N. Rate | RMSD |
|---|---|---|---|---|
| Linear | ODE | None: $\gamma = 0$ | 18.59% | 0.2939 |
| Linear | ODE | LatentSqrt: $\gamma = \sqrt{1.450\,t\,(1-t)}$ | 9.95% | 1.6660 |
| Linear | SDE | LatentSqrt: $\gamma = \sqrt{0.018\,t\,(1-t)}$ | 22.07% | 0.6148 |
| Trigonometric | ODE | None: $\gamma = 0$ | 19.63% | 0.8289 |
| Trigonometric | ODE | LatentSqrt: $\gamma = \sqrt{0.027\,t\,(1-t)}$ | 19.96% | 0.6570 |
| Trigonometric | SDE | LatentSqrt: $\gamma = \sqrt{0.023\,t\,(1-t)}$ | 17.60% | 0.7763 |
| Encoder-Decoder | ODE | Enc-Dec: $\gamma = \sin^2(\pi t)$ | 17.59% | 0.3899 |
| Encoder-Decoder | SDE | Enc-Dec: $\gamma = \sqrt{0.10}\,\sin^2\left(\frac{\pi(t-0.73t)^{0.5}}{(0.73-0.73t)^{0.5}+(t-0.73t)^{0.5}}\right)$ | 16.27% | 1.1795 |
| VP Score-Based Diffusion | ODE | $\sigma_0 = 0.23$ | 17.30% | 1.1376 |
| VP Score-Based Diffusion | SDE | $\sigma_0 = 7.14$ | 22.10% | 0.7631 |
| VE Score-Based Diffusion | ODE | $\sigma_0 = 0.45$; $\sigma_{\min} = 0.0021$, $\sigma_{\max} = 0.8319$ | 20.38% | 0.6644 |

*Table 11.* Study for the *MP-20* dataset comparing DNG hyperparameters for choice of positional interpolant, sample scheme, and $\gamma(t)$ (as reported in Tab. 10).

| Pos. interpolant, Sampling scheme, $\gamma$ | Cell interpolant Sampling scheme, $\gamma(t)$ | Annealing param. $s$ (Pos. / Cell) | Integration steps | Min. dist. permutation | Niggli | Species noise $\eta$ | $\lambda_{\boldsymbol{x},b}/\lambda_{\boldsymbol{l},b}/\lambda_{\boldsymbol{x},z}/\lambda_{\boldsymbol{l},z}/\lambda_{\boldsymbol{a}}$ |
|---|---|---|---|---|---|---|---|
| Linear, ODE, None | Linear, ODE, $\gamma = 0$ | 13.62 / 1.07 | 150 | True | False | 7.08 | 0.9775 / 0.0006 / - / - / 0.0218 |
| Linear, ODE, LatentSqrt | Enc-Dec, SDE, $\gamma = \sqrt{7.88}\,\sin^2\left(\frac{\pi(t-0.14t)}{(0.14-0.14t)+(t-0.14t)}\right)$ | 14.83 / 5.91 | 130 | True | False | 23.87 | 0.7683 / 0.0089 / - / 0.0012 / 0.2216 |
| Linear, SDE, LatentSqrt | Linear, ODE, $\gamma = 0$ | 6.33 / 1.07 | 710 | True | False | 0.19 | 0.1309 / 0.0065 / 0.2708 / - / 0.5918 |
| Trig, ODE, None | Trig, ODE, $\gamma = \sqrt{1.183\,t\,(1-t)}$ | 8.59 / 0.29 | 860 | True | False | 32.69 | 0.3302 / 0.0023 / - / - / 0.6675 |
| Trig, ODE, LatentSqrt | Linear, SDE, $\gamma = \sqrt{0.848\,t\,(1-t)}$ | 7.79 / 0.30 | 680 | True | True | 27.25 | 0.2322 / 0.0035 / - / 0.3338 / 0.4306 |
| Trig, SDE, LatentSqrt | Trig, ODE, $\gamma = \sqrt{0.316\,t\,(1-t)}$ | 12.80 / 4.36 | 760 | True | False | 13.15 | 0.6304 / 0.1582 / 0.0753 / - / 0.1360 |
| Enc-Dec, ODE, Enc-Dec | Linear, ODE, $\gamma = 0$ | 10.27 / 0.08 | 840 | False | False | 0.85 | 0.7268 / 0.0084 / - / - / 0.2648 |
| Enc-Dec, SDE, Enc-Dec | Linear, ODE, $\gamma = \sqrt{1.651\,t\,(1-t)}$ | 7.87 / 3.92 | 610 | False | False | 19.78 | 0.2143 / 0.1547 / 0.1968 / - / 0.4341 |
| VP SBD, ODE | Trig, ODE, $\gamma = \sqrt{7.797\,t\,(1-t)}$ | 2.30 / 2.74 | 710 | False | False | 20.27 | 0.4053 / 0.0447 / - / - / 0.5500 |
| VP SBD, SDE | Trig, SDE, $\gamma = \sqrt{3.100\,t\,(1-t)}$ | 9.06 / 11.77 | 870 | False | False | 8.52 | 0.5184 / 0.0044 / 0.0008 / 0.1180 / 0.3584 |
| VE SBD, ODE | Linear, SDE, $\gamma = \sqrt{0.913\,t\,(1-t)}$ | 12.72 / 0.98 | 330 | False | True | 5.87 | 0.2209 / 0.0430 / - / 0.6371 / 0.0990 |

# D. Evaluation Metrics

In this section, we provide details and discussion of the various metrics we use to evaluate CSP and DNG models (see Section 5.1).

## D.1. Match Rate and RMSE

The tradeoff between match rate and RMSE most strongly influences the *perov-5* dataset. We show in Fig. 7 how different positional interpolants for the atomic coordinates (trigonometric *vs.* linear with ODE sampling schemes) learn to generate matched structures differently. For the linear case, the change in matching tolerance (*via* the `ltol` parameter of Pymatgen's `StructureMatcher`) makes little difference. For the trigonometric interpolant, it makes a far more significant difference and leads to a much higher match rate, suggesting that the trigonometric interpolant learns structures more reliably but less accurately.

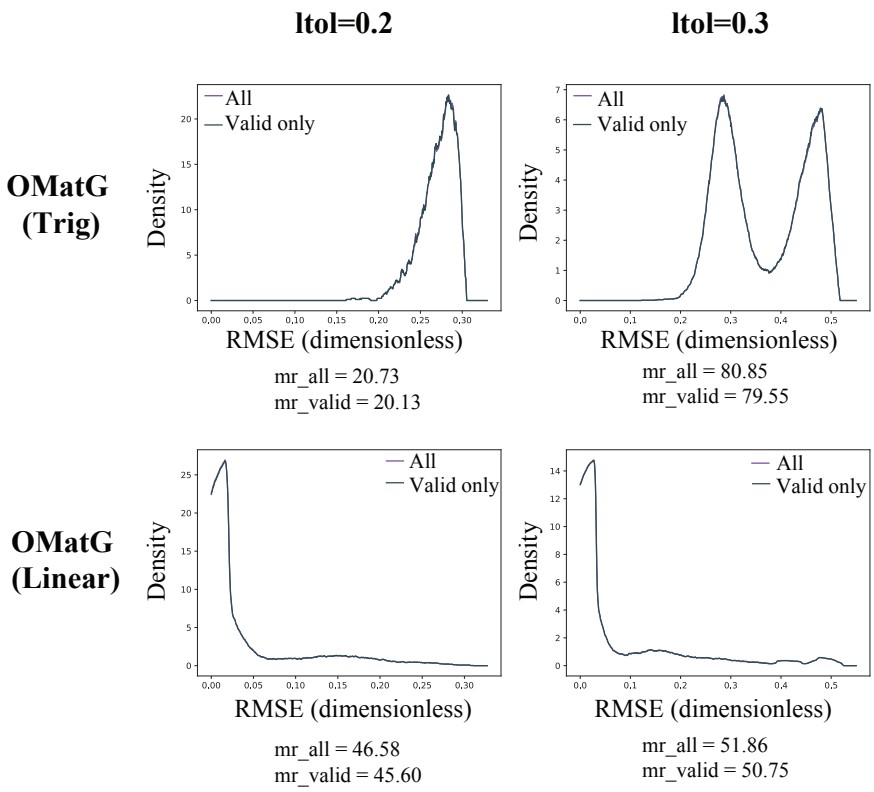

*Figure 7.* We show here the effect of making matching more difficult by decreasing the length tolerance used by Pymatgen's `StructureMatcher`. We plot the density of the normalized RMSE distributions from CSP models trained on the *perov-5* dataset (Linear and Trigonometric positional interpolants with ODE sampling schemes). We note that the curves for all generated structures and only valid generated structures overlap significantly.

## D.2. Validity Metrics

The structural validity of generated structures is defined according to the bond lengths present in the structure—all lengths must be >0.5 Å to be considered valid. The compositional validity is defined according to the SMACT software package (Davies et al., 2019). We note that the default oxidation states have been updated with the release of SMACT version 3.0 which changed the DNG compositional validity rates by several percent. This also impacts the CSP match rate when filtered by valid structures. As such, all values for all models were recomputed with the most up-to-date version (3.0) of the SMACT software.

## D.3. Coverage and Property Statistics

As in (Xie et al., 2022), we evaluate coverage recall and precision (reported as rates) by measuring the percentage of crystals in the test set and in the generated samples that match each other within a defined fingerprint distance threshold. For structural matching, we use CrystalNN, and for compositional matching, we use Magpie fingerprints. Additionally, we report Wasserstein distances between property distributions of the generated and reference datasets. The considered properties are the mass density ($\rho$), the number of unique elements ($N$ary), and the average coordination number for each element in the unit cell ($\langle CN \rangle$).

## D.4. Calculation of S.U.N. Rates

Evaluation of DNG structures was performed using scripts provided by the developers of MatterGen (MatterGen, 2025). A total of 10,000 structures were generated from each of OMatG, DiffCSP, FlowMM, and MatterGen-MP. These structures were then filtered to remove any that contained elements not supported by the MatterSim potential (version `MatterSim-v1.0.0-1M`) (Yang et al., 2024) or the reference convex hull. These included heavy elements with atomic numbers >89, radioactive elements, and the noble gases (specifically: 'Ac', 'U', 'Th', 'Ne', 'Tc', 'Kr', 'Pu', 'Np', 'Xe', 'Pm', 'He', 'Pa').[7] Stability and novelty were computed with respect to the default dataset provided by MatterGen which contains $845\,997$ structures from the *MP-20* (Jain et al., 2013; Xie et al., 2022) and *Alexandria* (Schmidt et al., 2022b;a) datasets. This provides a more challenging reference for computing novelty as each model was trained only on the $\sim 27\,000$ structures from the *MP-20* training set.

The MatterSim potential was utilized for the first structural relaxation, requiring far less compute resources compared to full DFT. Results derived from the MatterSim-relaxed structures are shown in Tab. 12. Following the MatterSim relaxtions, 1000 structures were relaxed with DFT. All DFT relaxations utilized `MPGGADoubleRelaxStatic` flows from the Atomate2 (Ganose et al., 2025) package to produce MP20-compatible data.

Comparing the results from Tab. 12 to Tab. 3, we find that overall there is reasonable agreement between the metrics computed at the machine learning potential and DFT level. Relative performance ordering between methods remains fairly consistent, allowing for qualitative trends to be made at the much cheaper ML potential level. Nevertheless, for a full quantitative understanding, DFT is essential.

*Table 12.* Stability (defined as $\leq 0.1$ eV/atom above hull), uniqueness, and novelty results from *de novo* generation on the MP-20 dataset computed for the same models as in Tab. 2. All evaluations are performed with the MatterGen code base (MatterGen, 2025) with respect to the included reference *Alex-MP-20* dataset. The average RMSD is between the generated and the relaxed structures, and the average energy above hull is reported in units of eV/atom. All structures were relaxed with the MatterSim potential. Note that the results in Tab. 3 relied on a subsequent DFT relaxation.

| Method | $\langle E \rangle/N$ ($\downarrow$) above hull | RMSD (Å, $\downarrow$) | Novelty Rate (%, $\uparrow$) | Stability Rate (%, $\uparrow$) | S.U.N. Rate (%, $\uparrow$) |
|---|---|---|---|---|---|
| DiffCSP | 0.1984 | 0.367 | 72.73 | 43.04 | 19.00 |
| FlowMM | 0.2509 | 0.651 | 72.76 | 37.47 | 13.86 |
| MatterGen-MP | 0.1724 | **0.142** | 72.17 | 47.07 | **22.66** |
| OMatG | | | | | |
| Linear | 0.1823 | 0.615 | 72.00 | 45.00 | 22.07 |
| Trig | 0.1857 | 0.657 | 65.35 | **51.40** | 19.96 |
| Enc-Dec | **0.1699** | 0.390 | 54.97 | **58.56** | 17.59 |
| SBD | 0.2189 | 0.763 | **75.80** | 42.60 | 22.10 |
| CFP + CSP | 0.2340 | 0.488 | **75.85** | 42.21 | 20.50 |

## D.5. Stability and Structural Analysis of Generated Structures

In Fig. 8, we show the distribution of computed energies above the convex hull across various OMatG models, showing best stability of generated structures for linear, encoder-decoder, trigonometric, and VP SBD positional interpolants.

---

[7]These elements were not removed from any datasets during training.

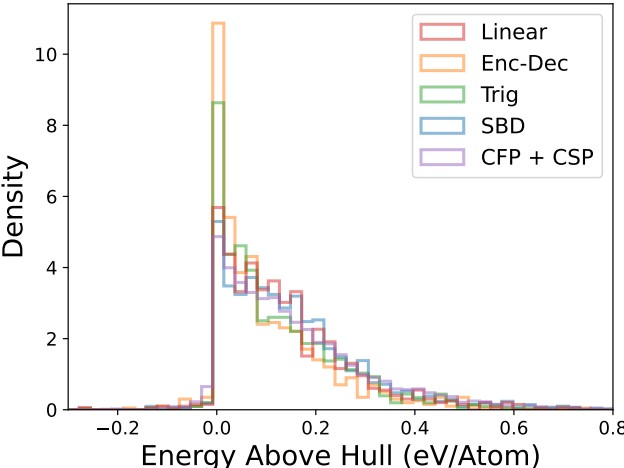

*Figure 8.* Histogram of the computed energies above the convex hull for structures generated by the five OMatG DNG models highlighted in the main text (see Table 3). We show that all positional interpolants are effective at generating structures close to the convex hull, with the VP SBD interpolant and CFP + CSP method performing slightly worse than other interpolants.

By evaluating the distribution of $N$ary structures (Fig. 9), the distribution of average coordination numbers (both by structure in Fig. 10 and by species in Fig. 11), as well as distribution of crystal systems (Fig. 12) which are related to a structure's Bravais lattice, we provide qualitative analysis for model performance. Space groups (and thus crystal systems) were determined using the `spglib` software (Togo et al., 2024) and choosing the most *common* space group identification with exponentially (geometrically) decreasing tolerance. We find that OMatG models have superior performance on the $N$ary metric across the board, with DiffCSP performing the most poorly. Specific OMatG models (with positional Encoder-Decoder interpolant and CFP+CSP with Linear interpolant) and DiffCSP showed the best performance in matching the distribution of average coordination number for each structure, particularly for high-coordinated structures. The average coordination number for species were best-matched by OMatG models across the board (with the exception of the OMatG-VPSBD model which tended to overpredict the coordination environments), and broadly underpredicted by DiffCSP, FlowMM, and MatterGen-MP. Finally, the best matching results for distribution across crystal systems was for the OMatG CFP+CSP model, with all other interpolants (and DiffCSP and FlowMM) showing a propensity for generating low-symmetry (triclinic and monoclinic) crystal structures. Overall, we find that OMatG models closely reproduce the elemental and structural diversity present in the data.

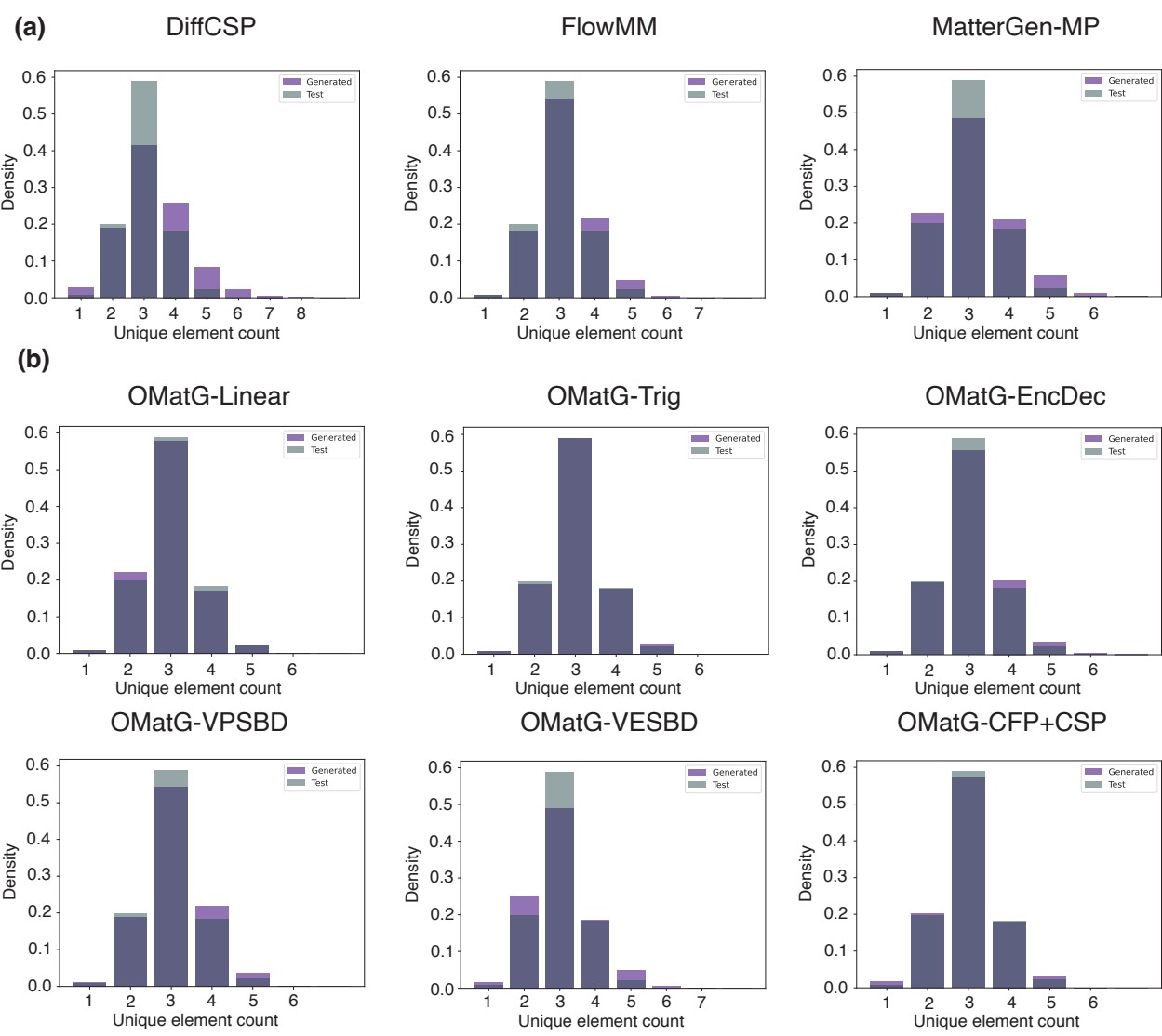

*Figure 9.* Qualitative performance of the distribution of $N$ary crystals for **(a)** Non-OMatG models and **(b)** OMatG models across structural benchmarks computed on generated structures and test set structures from the *MP-20* dataset. Atomic elements are listed in increasing atomic number from left to right.

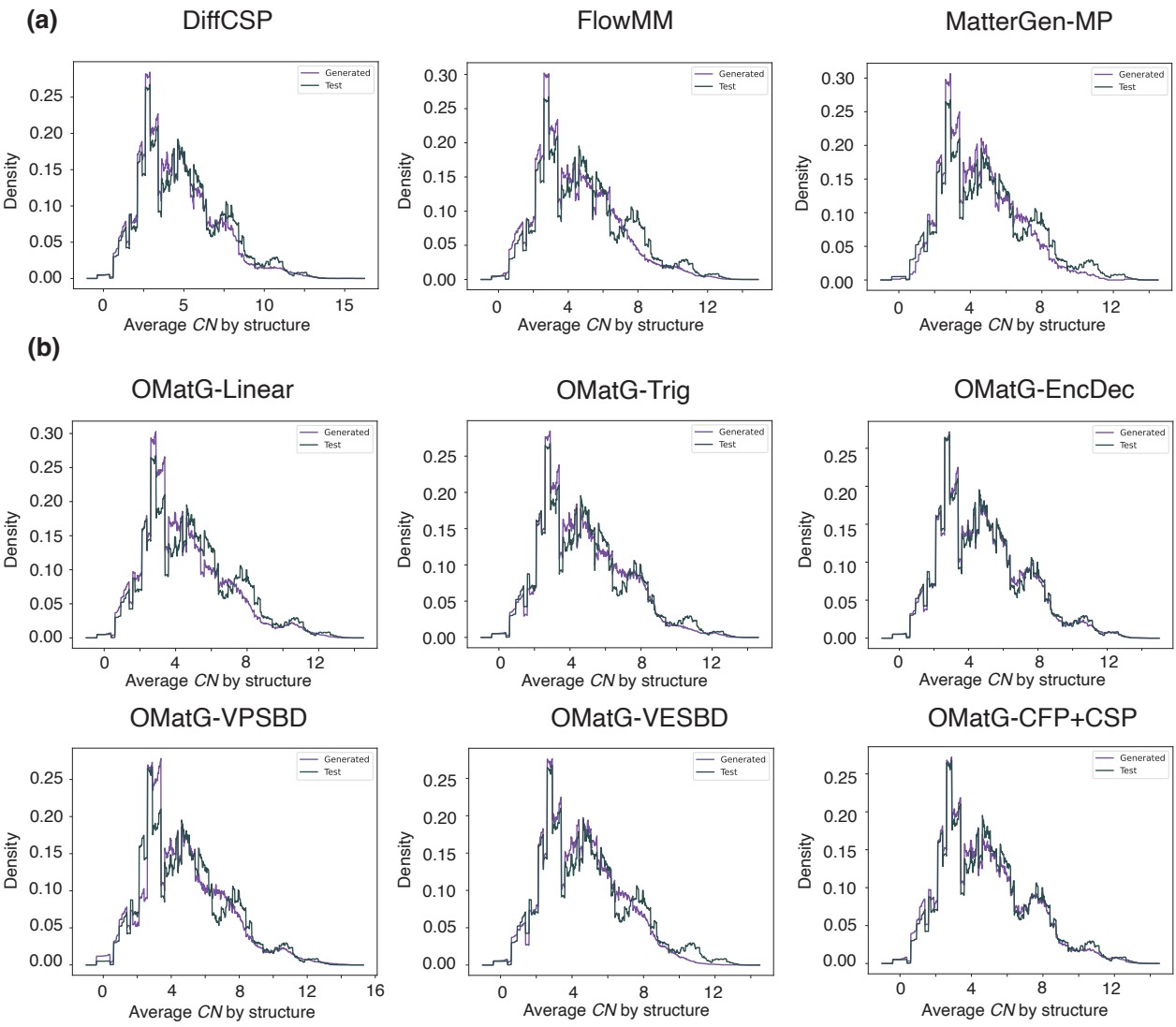

*Figure 10.* Qualitative performance of the distribution of average coordination number by structure for **(a)** Non-OMatG models and **(b)** OMatG models across structural benchmarks computed on generated structures and test set structures from the *MP-20* dataset.

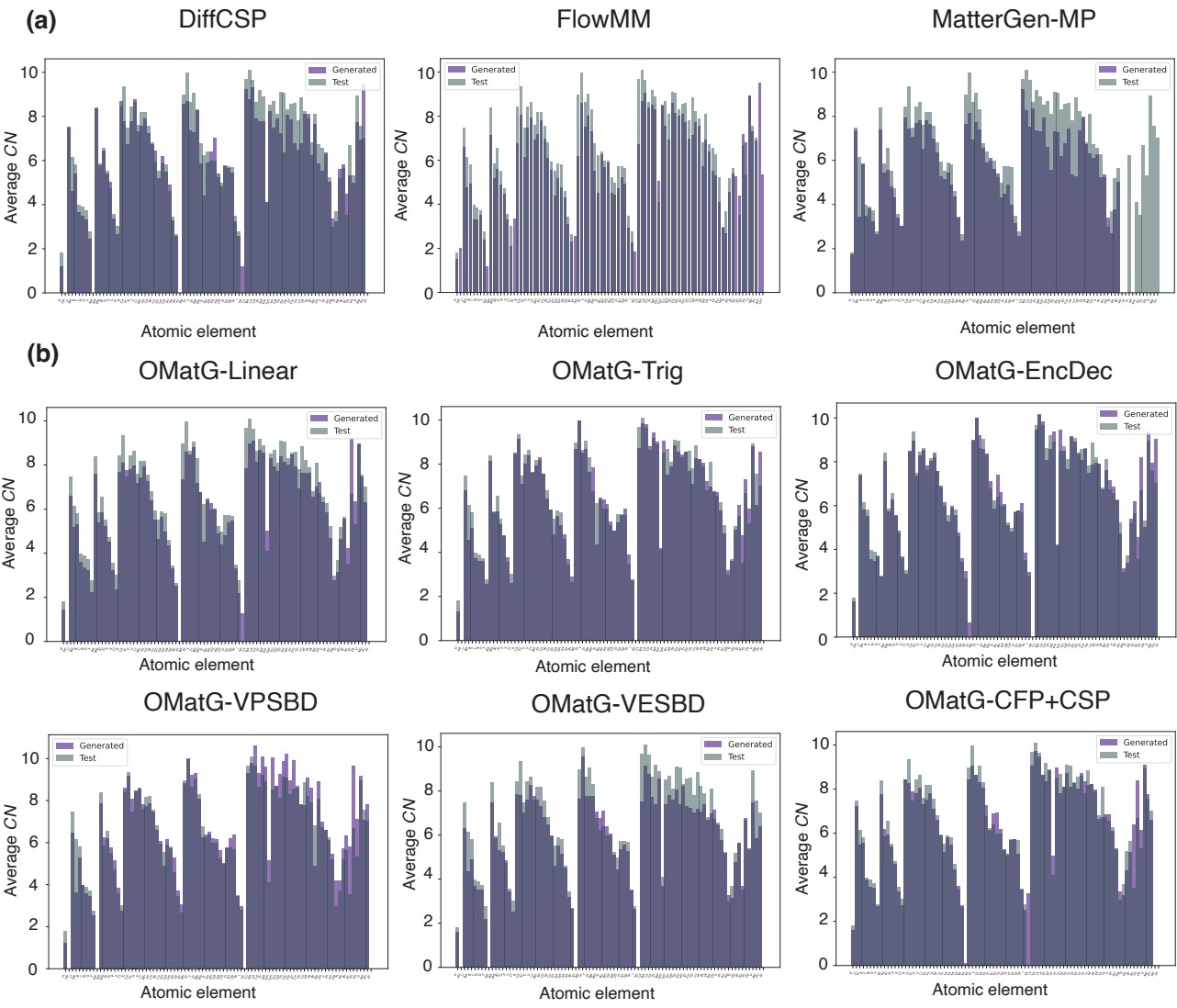

*Figure 11.* Qualitative performance of the distribution of average coordination number by species (listed left to right in order of atomic number) for **(a)** Non-OMatG models and **(b)** OMatG models across structural benchmarks computed on generated structures and test set structures from the *MP-20* dataset.

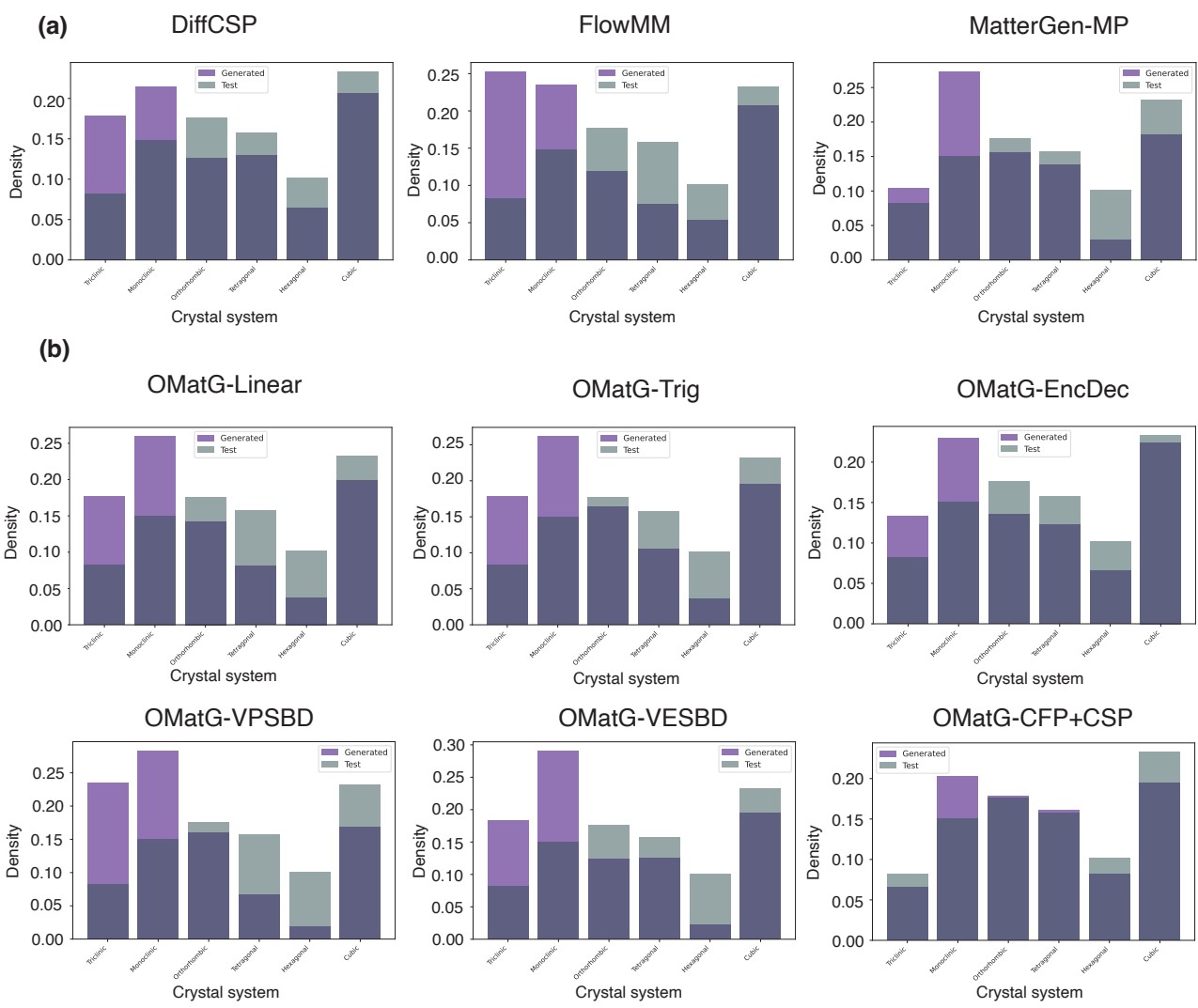

*Figure 12.* Qualitative performance of the distribution of crystal system by structure for **(a)** Non-OMatG models and **(b)** OMatG models across structural benchmarks computed on generated structures and test set structures from the *MP-20* dataset.

# E. Large Language Models as Base Distributions

FlowLLM (Sriram et al., 2024) combines large language models (LLMs) with the conditional flow-matching framework FlowMM (Miller et al., 2024) to design novel crystalline materials in the DNG task. A fine-tuned LLM serves as the base distribution and samples initial structures; FlowMM then refines the fractional coordinates and lattice parameters as in the CSP task. This idea can be similarly applied to OMatG, which allows the use of LLMs within the general SI framework for materials generation.

We extend OMatG to OMatG-LLM by allowing for LLM-generated structures as the initial structures. We evaluate both FlowLLM and OMatG-LLM on the LLM dataset released by FlowLLM. Specifically, we use the training (containing 40 000 structures) and validation sets (6000 structures) from `https://github.com/facebookresearch/flowmm` and the LLM-generated initial structures (10 000 structures) from `https://github.com/facebookresearch/crystal-text-llm` as the test set. These initial structures are generated by a fine-tuned Llama-70B model (Gruver et al., 2024). As shown in Tab. 13, OMatG-LLM's linear and trigonometric interpolants outperform FlowLLM in almost all DNG metrics. Since the Wasserstein distance with respect to the $N$arity distributions and the compositional validity only depend on the atomic species generated by the LLM, these two metrics are necessarily equal for FlowLLM and OMatG-LLM. Note that the original FlowLLM (Sriram et al., 2024) is trained on 3M LLM-generated structures while our experiments are conducted on the 40K structures open-sourced by the authors. The performance of FlowLLM in our experiments thus differs from the scores reported in (Sriram et al., 2024).

*Table 13.* FlowLLM's and OMatG-LLM's performance (with linear and the trigonometric interpolants) when using the same fine-tuned LLM (Gruver et al., 2024) as the base distribution. The best performance for each metric is in bold.

| Method | Validity (% ↑) | | | Coverage (% ↑) | | Property (↓) | | | S.U.N. Rate (% ↑) |
|---|---|---|---|---|---|---|---|---|---|
| | Structural | Composition | Combined | Recall | Precision | wdist ($\rho$) | wdist ($N$ary) | wdist ($\langle CN \rangle$) | |
| FlowLLM | 96.27 | 86.40 | 83.55 | 97.98 | 96.55 | 0.9922 | 0.5427 | 0.5936 | 10.28 |
| OMatG-LLM | | | | | | | | | |
| Linear | **97.86** | 86.40 | **84.85** | **99.16** | **98.40** | 0.9100 | 0.5427 | 0.8600 | **12.61** |
| Trigonometric (ODE) | 95.70 | 86.40 | 83.25 | **98.57** | 98.24 | **0.7410** | 0.5427 | 0.6165 | **11.14** |
| Trigonometric (SDE) | **97.78** | 86.40 | **84.72** | 97.41 | **99.12** | 3.6214 | 0.5427 | **0.4448** | **11.86** |

# F. Computational Costs

In Tabs 14 and 15, we present the computational costs for both the CSP and DNG tasks. We compare the cost of training and integrating OMatG on the *MP*-20 dataset and show low computational costs for OMatG's ODE scheme for both training and inference. The SDE scheme is more expensive but competitive. For these experiments, we use an Nvidia RTX8000 GPU with a batch size of 512 and 1000 integration steps.

*Table 14.* Computational costs for DiffCSP, FlowMM, OMatG (ODE) and OMatG (SDE) models trained on the CSP task.

| Task | OMatG (ODE) | FlowMM | OMatG (SDE) | DiffCSP |
|---|---|---|---|---|
| Training (s / epoch) | $56.8 \pm 0.75$ | $70.35 \pm 1.38$ | $89.0 \pm 1.41$ | $21.89 \pm 0.31$ |
| Sampling (s / batch) | $313.67 \pm 9.29$ | $424.125 \pm 11.78$ | $479.5 \pm 13.5$ | $338.11 \pm 11.93$ |

*Table 15.* Computational costs for DiffCSP, FlowMM, OMatG (ODE) and OMatG (SDE) models trained on the DNG task.

| Task | OMatG (ODE) | FlowMM | OMatG (SDE) | DiffCSP |
|---|---|---|---|---|
| Training (s / epoch) | $75.26 \pm 2.08$ | $73.32 \pm 0.47$ | $102.65 \pm 1.87$ | $21.85 \pm 0.36$ |
| Sampling (s / batch) | $473.14 \pm 13.20$ | $469.93 \pm 6.12$ | $617.2 \pm 18.2$ | $322.63 \pm 10.28$ |

