# OpenReview forum: "Open Materials Generation with Stochastic Interpolants"
_ICML.cc/2025/Conference — ICML 2025 poster_

### Official Review · Reviewer_K1be · 2025-03-13

**Overall Recommendation:** 4

**Summary:**

The paper presents an extension of stochastic interpolants for the modelling of crystalline materials. Stochastic interpolants are a general framework that encompasses diffusion models and flow matching as specific instances. As the fractional coordinates live on a torus, they adapt the interpolants to respect the circular nature of the space. They use stochastic interpolants also to model the unit cell parameters (lengths and angles), while they rely on discrete flow matching in the case of atom types. They present results for two different tasks: crystal structure prediction (CSP) and de novo generation (DNG).


### After rebuttal period
I think that the author's rebuttal clarified the minor concerns I had and the additional results make the paper even stronger.

**Claims And Evidence:**

The claims that the paper presents are that stochastic interpolants can be used for crystalline material generation and that they outperform all the other deep generative model approaches in the literature. While the first claim is well supported, the second one is a bit tricky as although they are better from the experiments table than the other methods, this is not an apple-to-apple comparison. Therefore, I think that more analysis is required to support this claim (see weaknesses)

**Essential References Not Discussed:**

/

**Experimental Designs Or Analyses:**

The experimental setup follows closely the ones of CDVAE and DiffCSP, therefore these are the classic experiments considered in papers for material generation.

**Methods And Evaluation Criteria:**

The benchmark and evaluation metrics are the ones that are usually considered in this context of generative models for materials. They also evaluate DNG samples using a foundation model, i.e. Mattergen (FlowMM used a different one but they recomputed results for that baseline using Mattergen). Although they are not evaluating with DFT as done in FlowMM, I think that this evaluation is enough to compare the different methods.

**Other Comments Or Suggestions:**

/

**Other Strengths And Weaknesses:**

The paper is well-written and easy to follow. Also, the evaluation of the unrelaxed DNG samples in terms of average energy above the hull, stability, uniqueness, and novelty using MatterGen as a foundation model strengthens the results. The main weakness is that comparison with previous models is not exactly apple-to-apple and therefore it is difficult to understand which ingredients are making the approach better. I think the paper will benefit from more ablation studies. For example, in the DNG task, do the improvements come from the use of stochastic interpolants for fractional coordinates and lattice parameters or mostly from the discrete flow matching approach for the atom types? Indeed, FLowMM uses analog bits and continuous flow matching. In addition to that, they propose different interpolants (both stochastic and deterministic), but present results only for the best approach, making it difficult to get the full picture of the design space.

**Questions For Authors:**

I have a few questions regarding some parts of the paper that I personally think would be nice to be discussed in the paper:

- The results on perov-5 make me a bit confused. I appreciate the discussion in the appendix on the sensitivity of the trigonometric interpolants, but at the same time, it is impressive how the tolerance affects the match rate for that specific interpolant in contrast to the linear one. Also by looking at the distribution over the RMSD it seems that the error is x3 bigger than the linear interpolant. It would also be interesting to get the full picture of how the trigonometric interpolant is performing in the other dataset. Is there something that can be learned from the experiment on perov-5 to use on the other dataset to get approximately the same improvement? Also, it seems that the trigonometric interpolant is the only one (from our experiments) that works better with $\gamma(t)$, is there a reason why in this case the model benefits more from this stochasticity at training time? It would be helpful to see the results of all the interpolants on perov-5 and MP-20 to see how also the increase in the number of atoms affects the performance.

- I am a bit confused when on line 181, you say that the base distribution for the score-based diffusion interpolants needs to be a wrapped normal distribution as in DiffCSP. In DiffCSP the wrapped normal is used to define the transition kernel $p_{t|0}(x_t|x_0)$, but the diffusion converges to a uniform distribution also in that case. I would be happy if you could comment on that sentence a bit more.

- It seems that there are plenty of hyperparameters to tune to get the final model. How many models have you trained for each dataset? I think it would be helpful to mention in the appendix all the ranges considered for every hyperparameter.

- In Figure 5 in the appendix, I am a bit confused as to why you need to first compute a geodesic before computing the interpolant. Can you elaborate more on this? Also, a geodesic on a torus is also an interpolant, does that correspond to the wrapped linear interpolant (with wrapped I mean the way you are computing interpolations as described in section 3.2.1) in your case?

- I know that removing the Euclidean mean from the target is done in FlowMM too, but is it enough to get a consistent training target for the network?

- Just to clarify: are you training your model on the usual MP20 dataset used also in DiffCSP and just doing the filtering you mention in the appendix (page 17) after sampling?

**Relation To Broader Scientific Literature:**

The paper extends Stochastic interpolants in the context of crystalline material generation. It presents a discussion of the main deep generative approaches used in the context of crystalline material generation.

**Theoretical Claims:**

-

---

> ### Author Rebuttal · Authors · 2025-04-01
>
> We appreciate the reviewer’s thorough engagement with our manuscript and thank them for their insightful feedback.
>
> ## Ablation studies, hyperparameters, and model performance
>
> Regarding the performance of different stochastic interpolants, we direct the reviewer to the CSP ablation study tables provided in response to Reviewer w22y in which we highlight the best-performing models for each unique combination of positional interpolant, sampling scheme, and latent variable.
>
> For the perov-5 ablation study, we show across the board that all interpolants but the linear one can beat the state-of-the-art match rate set previously by DiffCSP/FlowMM’s models, all with larger RMSEs (our linear interpolant performs comparably to both models). The increased RMSE is partly why the match rate increases: for SIs outside the linear interpolants, we find that particles generally find the correct local chemical configurations to flow towards, but are not able to end up in the precise symmetric sites. By contrast, the linear interpolants have the lowest RMSE because the particles flow to more symmetric positions, but the local environments are not correct due to species mismatch. We note that the trigonometric interpolant is not unique in its ability to have a high match rate. We suspect that the encoder’s ability to learn relevant representations for species may pose a limitation, which noised or non-geodesic interpolating paths can overcome. We can **add this revised discussion to the main text.**
>
> We also show that for MP-20, the trigonometric interpolant with an SDE can also outperform on match rate compared to previously published models.
>
> Comparing the perov-5 and MP-20 datasets to understand the **effect of unit cell size** would _not_ be effective since these datasets have vastly different atomic, species, and unit cell distributions. The comparison between MP-20 and MPTS-52 would be more pertinent as they are more similar: they are both taken from the MP database, and differ by the max. number of atoms (20 vs. 52). Their match rates are reported in the original manuscript.
>
> Concerning the number of trained models: Many models were _partially_ trained and compared in the process of hyperparameter tuning: on average 27 models (perov-5) and 32 models (MP-20) for each choice of positional interpolant, sample scheme, and latent variable. **We will add ranges for the hyperparameters to the appendix.**
>
> ## Comparison to FlowMM
>
> The notable differences between our models and FlowMM are (a) **discrete flow matching on species** for OMG vs. analog bits for FlowMM; (b) the **cell representation**; (c) FlowMM’s use of a **slightly modified CSPNet encoder** while OMG utilizes CSPNet out of the box. We direct the reviewer to our CSP results, which show improvement over the FlowMM’s model we train **without any species learning.** Thus the handling of species is not sufficient to fully explain the differences in model performance.
>
> ## Clarifications
>
> ### SBD base distribution
>
> We agree with the reviewer that the wrapped normal distribution with a large variance as used in DiffCSP can be approximated by a uniform distribution. We made the referenced note in line 181 to reflect our implementation and to highlight the connection to one-sided interpolants in the SI framework (that require a normal base distribution). **We will update the corresponding sentence to clarify this.**
>
> ### Geodesic and periodic interpolants
>
> The geodesic is indeed the same as the linear interpolant wrapped back into the box. The reason for computing the geodesic first for all other interpolants is because there are multiple ways to connect two points on a torus (e.g., in a periodic box one can connect two points with or without crossing the box boundaries). Using the geodesic as the “starting point” for computing the interpolating path allows for them to be uniquely defined. We also point to our response to Reviewer gm5E in section “PBCs.”
>
> ### Subtraction of COM motion
>
> The removal of the center-of-mass motion (as similarly implemented by FlowMM) in the loss function is basically analogous to choosing translationally invariant representations of the unit cells (see the discussion in Appendix D of Miller et al., 2024; arXiv:2406.04713). This allows to train of the translationally invariant CSPNet model in a consistent manner. Phrased differently, CSPNet cannot predict any COM motion which is why this part has to be removed in the ground-truth velocity for a consistent training target.
>
> ### Species filtering
>
> Our models are indeed being trained on the full MP-20 dataset (with all atom types) and the filtering of atoms is only done during relaxation with MatterSim. **We will highlight this more clearly in the revised manuscript.**

---

### Official Review · Reviewer_w22y · 2025-03-14

**Overall Recommendation:** 3

**Summary:**

The paper introduced Open Materials Generation (OMG), a framework that leverages stochastic interpolants in generative models for inorganic crystalline materials. The method is built on existing architecture in the literature (CSPNet), which is based on an equivariant graph neural network (EGNN). The authors addressed two materials tasks: Crystal structure prediction for fixed compositions and de novo generation. The method has been evaluated on two materials datasets: perov-5 and MP-20.

**Claims And Evidence:**

The authors claimed to achieve state-of-the-art but I think there is a reference missing (see the Essential References section).

**Essential References Not Discussed:**

The following reference is missing in the comparison and the method used the same benchmark:

- FlowLLM: Flow Matching for Material Generation with Large Language Models as Base Distributions. Sriram et al., 2024.

**Experimental Designs Or Analyses:**

Yes.

**Methods And Evaluation Criteria:**

Yes.

**Other Comments Or Suggestions:**

No.

**Other Strengths And Weaknesses:**

**Strengths**:

- Integration of stochastic interpolants: I think the idea of extending stochastic interpolants to generate materials is novel.
- Experimental results: The paper showed strong empirical results of the proposed method on materials datasets.

**Weaknesses**:

- Computational requirements: The paper did not mention the training or inference costs of the proposed method, or how it compares to previous approaches.
- Limited baselines: As mentioned by the authors, the paper did not compare with symmetry-aware models (like Crystal-GFN or  WyCryst) which narrows the scope of the proposed method.

**Questions For Authors:**

Can you explain more about the choice and limitations of each interpolant, and how this affects the performance across different datasets?

**Relation To Broader Scientific Literature:**

The method is built on the existing approaches in the literature like CSPNet architecture and stochastic interpolants.

**Theoretical Claims:**

No.

---

> ### Author Rebuttal · Authors · 2025-04-01
>
> We thank the reviewer for raising questions about our paper, and address them topically below.
>
> ## Comparison with FlowLLM
> We agree that FlowLLM is an important material generation method representing the most recent trends. It uses an LLM to sample structures and FlowMM to refine them. We note that this is an orthogonal feature to our method which uses the general SI framework, and that FlowLLM’s approach can be incorporated into OMG. For a fair comparison, we evaluate both FlowMM (FlowLLM) and OMG (OMG-LLM) on the LLM dataset released by the FlowLLM authors (see https://github.com/facebookresearch/flowmm) and show results below.
>
> || LLM model size | Cov. precision | Cov. recall | wdist $\rho$ | wdist $\langle CN \rangle$ | Structural validity |
> | --- | --- | --- | --- | --- | --- | --- |
> FlowLLM | 70B | 96.55 | 97.98 | 0.9922 | 0.5936 | 96.27 |
> OMG-LLM | 70B | 98.40 | 99.16 | 0.9100 | 0.8600 | 97.86 |
>
> Refining the same structures generated by an LLM, OMG’s linear interpolant outperforms FlowMM in almost all DNG metrics. We appreciate the reviewer’s suggestion as this comparison–**to be included in the revised paper**–further demonstrates the advantage and flexibility of OMG.
>
> ## Computational cost
> We compare the cost of training and integrating OMG on the MP-20 dataset and **show low computational costs for OMG’s ODE scheme for both training and inference**. The SDE scheme is more expensive but competitive. For these experiments, we use an Nvidia RTX8000 GPU with a batch size of 512 and 1000 integration steps. **These results will be included in the appendix.**
> ### CSP
> |   Task   | OMG (ODE) | FlowMM | OMG (SDE) | DiffCSP |
> |-----------|-------------|--------|-----------|---------|
> | Training (s / epoch) |     $56.8 \pm 0.75$     |    $70.35 \pm 1.38$   |     $89.0 \pm 1.41$     |    $21.89 \pm 0.31$    |
> | Sampling (s / batch) |     $313.67 \pm 9.29 $     |    $424.125 \pm 11.78$   |     $479.5 \pm 13.5$     |    $338.11 \pm 11.93$    |
>
> ### DNG
> |   Task   | OMG (ODE) | FlowMM | OMG (SDE) | DiffCSP |
> |-----------|-------------|--------|-----------|---------|
> | Training (s / epoch) |     $75.26 \pm 2.08$     |    $73.32 \pm 0.47$   |     $102.65 \pm 1.87$     |    $21.85 \pm 0.36$    |
> | Sampling (s / batch) |     $473.14 \pm 13.20$     |    $469.93 \pm 6.12$   |     $617.2 \pm 18.2$     |    $322.63 \pm 10.28$    |
>
> ## Ablation studies
> Regarding the performance across positional interpolants, we provide ablation studies for perov-5 and MP-20 on the CSP task, broken down by choice of positional interpolant, sampling method, and latent variable $\gamma$ (**to be added to the appendix**). We note different trends for the perov-5 dataset, which has cubic unit cells and similar positions, and the MP-20 dataset, which exhibits more structural and chemical variation. We direct the reviewer to our discussion in response to Reviewer K1be in section “Ablation studies, hyperparameters, and model performance”.
>
> **Perov-5 CSP**
> | Pos. Interpolant | Pos. sampling | Pos. gamma | Match rate (%, Valid only) | RMSE (Valid only) |
> | --- | --- | --- | --- | --- |
> | Linear | ODE | $\gamma(t)=0$ | 50.62 | **0.0760** |
> | Linear | ODE | $\gamma(t)=\sqrt{0.034t(1-t)}$ | **62.54** | 0.3444 |
> | Linear | SDE | $\gamma(t)=\sqrt{0.028t(1-t)}$ | **72.87** | 0.3315 |
> | Trig | ODE | $\gamma(t)=0$ | **52.36** | 0.3628 |
> | Trig | ODE | $\gamma(t)=\sqrt{0.011t(1-t)}$ | **79.55** | 0.3873 |
> | Trig | SDE | $\gamma(t)=\sqrt{0.063t(1-t)}$ | **71.60** | 0.3614 |
> | Enc-Dec | ODE | $\gamma(t)=\sqrt{0.66} \sin^2(\pi(t-0.80t) / ((0.80-0.80t) + (t - 0.80t)))$ | **64.60** | 0.4003 |
> | Enc-Dec | SDE | $\gamma(t)=\sqrt{8.45} \sin^2(\pi(t-0.61t) / ((0.61-0.61t) + (t - 0.61t)))$ | **76.80** |0.3620 |
> | SBD | ODE | $\sigma = 0.28$ | **81.27** | 0.3755 |
> | SBD | SDE | $\sigma = 0.13$ | **64.46** | 0.3402 |
>
> **MP-20 CSP**
> | Pos. Interpolant | Pos. sampling | Pos. gamma | Match rate (%, Valid only) | RMSE (Valid only) |
> | --- | --- | --- | --- | --- |
> | Linear | ODE | $\gamma(t)=0$ | **63.75** | 0.0720 |
> | Linear | ODE | $\gamma(t)=\sqrt{0.257t(1-t)}$ | 50.04 | 0.1494 |
> | Linear | SDE | $\gamma(t)=\sqrt{0.063t(1-t)}$ | **61.88** | 0.1611 |
> | Trig | ODE | $\gamma(t)=0$ | 58.94 | 0.1149 |
> | Trig | ODE | $\gamma(t)=\sqrt{0.033t(1-t)}$ | 59.15 | 0.0998 |
> | Trig | SDE | $\gamma(t)=\sqrt{0.049t(1-t)}$ | **61.39** | 0.1321 |
> | Enc-Dec | ODE | $\gamma(t) = \sqrt{1.99} * \sin^2(\pi(t - 0.65t) / ((0.65 - 0.65t) + (t - 0.65t)))$ | 49.45 | 0.1260 |
> | Enc-Dec | SDE | $\gamma(t) = \sqrt{0.04} * \sin^2(\pi(t - 0.42t)^{0.5} / ((0.42 - 0.42t)^{0.5} + (t - 0.42t)^{0.5}))$ | 52.44 | 0.1125 |
> | SBD | ODE | $\sigma=0.22$ | 37.39 | 0.1890 |
> | SBD | SDE | $\sigma=2.29$ | 38.08 | 0.2088 |
>
> ## Limited baselines
> We reiterate that symmetry-aware methods would not be an apples-to-apples comparison to our method, and thus are not utilized in benchmarks. However, they are discussed in the manuscript and can be incorporated into future iterations of OMG.

---

### Official Review · Reviewer_ENYj · 2025-03-14

**Overall Recommendation:** 3

**Summary:**

This paper introduces a framework called OMG that applies stochastic interpolants to generate inorganic crystalline materials. The authors adapt the stochastic interpolants framework to handle periodic boundary conditions for crystal structures and integrate discrete flow matching for atomic species. Their approach provides flexibility in choosing interpolation schemes and sampling methods, outperforming existing methods on CSP and de novo generation tasks.

**Claims And Evidence:**

The main claim that OMG achieves state-of-the-art performance on CSP and DNG tasks is supported by comprehensive benchmarking across multiple datasets. The authors demonstrate performance improvements over DiffCSP and FlowMM, and show comparable results with MatterGen. The authors also claims for the flexibility of their approach. I believe the claims can be supported by the ablation studies on how different interpolant choices optimize performance for different datasets and tasks.

**Essential References Not Discussed:**

N/A

**Experimental Designs Or Analyses:**

Experiments are comprehensive.

**Methods And Evaluation Criteria:**

The stability evaluation using MatterSim provides a computationally efficient alternative to DFT relaxations. However, I would still suggest DFT calculation for accurate and fair comparison on crystal structure stability evaluation.

**Other Comments Or Suggestions:**

N/A

**Other Strengths And Weaknesses:**

Weaknesses:
1. I suggest some more theoretical analysis explaining why specific interpolant choices work preferentially on each dataset.
2. While stability rate is reported, the approach is not validated by experimental synthesis of novel materials. DFT calculation or CHGNet should provide a more comprehensive comparison to other methods on quality and stability of the generated structures.

**Questions For Authors:**

1. Have you explored how the choice of interpolants affects the diversity of generated structures beyond the standard property distribution metrics, for example elemental distribution?
2. Are you evaluating structures without relaxation? Have you analyzed the discrepancy of your generated structures before and after relaxation?

**Relation To Broader Scientific Literature:**

The authors place their work appropriately within the context of both materials generation and generative modeling literature. They acknowledge the state-of-the-art in both fields and clearly articulate how their approach bridges these domains.

**Theoretical Claims:**

The integration with discrete flow matching for atomic species are novel.

---

> ### Author Rebuttal · Authors · 2025-04-01
>
> We thank the reviewer for their feedback and address their concerns below.
>
> ## DFT relaxation
> We agree that DFT relaxations offer a more rigorous evaluation of structure stability. As such, we are currently running DFT calculations for a large batch of generated structures. To assess consistency between the MLIP (MatterSim) and DFT results, we analyze 10 random subsets of 100 structures each (from the ~800 structures for which DFT relaxations are currently complete). For each subset, we compute the metastable S.U.N. (M.S.U.N.) rate based on the MLIP and DFT relaxed structures, respectively:
> | MLIP M.S.U.N. | DFT M.S.U.N. |
> |-------------|-----------|
> | 0.16        | 0.09      |
> | 0.13        | 0.12      |
> | 0.06        | 0.06      |
> | 0.12        | 0.10      |
> | 0.14        | 0.12      |
> | 0.13        | 0.12      |
> | 0.16        | 0.15      |
> | 0.10        | 0.09      |
> | 0.13        | 0.13      |
> | 0.11        | 0.08      |
>
> As the table shows, **we observe close agreement between the MLIP and DFT**, with DFT M.S.U.N. rates consistently tracking the MLIP rates while showing slightly more conservative values.
>
> To further validate consistency, we also compared the energy above the convex hull between MLIP and DFT-relaxed structures. We find strong agreement, with a linear regression producing $R^2 = 0.986$, indicating that the MLIP (MatterSim) serves as a reliable surrogate for DFT.
>
> **The complete results will be included in the revised version of the paper.**
>
> ## Interpolant choice and structural diversity
>
> We thank the reviewer for raising the important question of how the interpolant choice affects diversity in generated materials. We refer the reviewer to our response to Reviewer w22y for additional ablation studies for CSP (**to be added to appendix**), and to Reviewer K1be in section “Ablation studies, hyperparameters, and model performance” for a more detailed discussion of how interpolant choice affects performance (**to be added to main text**).
>
> For the DNG task on the MP-20 dataset, we have also obtained **$N$-ary distributions (i.e., number of unique elements per structure) and element-wise distributions of average coordination number** across all positional interpolants used in OMG. We find that the best OMG models show superior agreement between the test set and the generated structures for these elemental distributions, and thus conclude that OMG can closely reproduce the elemental and structural diversity present in the data. In particular, the OMG-Linear, OMG-EncDec, and OMG-CFP+CSP positional interpolants show best agreement for the $N$-ary distributions, and all OMG models show superior performance on the element-wise distributions of average coordination number where DiffCSP and FlowMM’s models show significant under-coordination of atomic environments. We will include these results as **new figures in the appendix**.
> ## Evaluation before and after relaxation
> As the reviewer correctly notes, evaluation can be performed either on generated structures as-is or after relaxation with DFT or an MLIP. Our evaluation is split accordingly:
> - Table 2 (main text) reports DNG performance _before_ relaxation, focusing on coverage (recall, precision), property distributions (e.g., density, average coordination number, $N$-ary count), and validity metrics (structural, compositional, and combined). These metrics are used to assess how well the model captures the target data distribution and should reflect the quality of generation prior to any refinement.
> - Table 3 (main text) reports DNG results _after_ structural relaxation using the MatterSim MLIP, evaluating stability, novelty and uniqueness, as well as RMSD between initial and relaxed structures. These metrics assess the model’s utility for materials discovery. The RMSD values directly quantify structural discrepancy between generation and relaxation. We discovered a transcription error in the initially reported values and provide the corrected RMSDs below:
>   - DiffCSP: 1.295
>   - FlowMM: 0.651
>   - OMG-Linear: **0.294**
>   - OMG-Trig: 0.763
>   - OMG-EncDec: **0.390**
>   - OMG-SBD: 0.759
>   - OMG-CFP+CSP: **0.488**
>
> These corrected values support OMG’s ability to generate structures that are not only diverse and realistic but also close to relaxed local minima, especially for the linear and encoder-decoder interpolants.

---

### Official Review · Reviewer_gm5E · 2025-03-23

**Overall Recommendation:** 3

**Summary:**

This paper extends flow-based inorganic crystalline structure prediction (CSP) to the stochastic interpolants (SI) framework. The authors use an equivariant graph representation (CSPNet) and wrapped interpolants to account for periodic boundary conditions of atomic coordinates and discrete flow-matching (DFM) to generate atomic species for De Novo Generation (DNG). By placing CSP into the SI framework, they are are able to show empirical performance gains over prior methods (DiffCSP, FlowMM, MatterGen) by ablating the over the additional tuning knobs offered by SI.

**Claims And Evidence:**

This paper claims that
-The SI framework is a unifying formulation that generalizes both flow matching and diffusion-based generative models.
-The method is flexible and tunable through the choice of interpolants and noise scheduling, contributing to better empirical results. Notably, they achieve state-of-the-art performance on both CSP and DNG tasks

Evidence:
The paper definitely supports its performance claims with comprehensive experimental comparisons across several datasets. Although the SI framework generally comes with more tuning knobs “out of the box”, conceptually it is mostly using the formulation of the previous flow-matching and diffusion works for material generation. With the right reparameterizations, the SI paths can be realized by flow matching probability paths.  Additionally, I think it's hard to argue that SI unifies FlowMM since it is based on Riemannian flow matching and intrinsically handles the PBC conditions (although I think in the case of flat tori, the geodesics proposed here for SI seem consistent).

**Essential References Not Discussed:**

I’m not aware of additional material generation references.

**Experimental Designs Or Analyses:**

The experimental design and analysis appears thorough. The authors compare against DiffCSP and FlowMM, reproducing their results accurately for several benchmarking. They also provide an informative ablations regarding hyperparameter tuning, interpolant choice.
It could be informative to also display computational cost associated with training and inference for OMG compared to existing methods.

**Methods And Evaluation Criteria:**

The SI framework are formulated pragmatically and well-motivated by CSP and DNG tasks. The evaluation criteria such as matching rate, RMSD and coverage are all reasonable metrics.

**Other Comments Or Suggestions:**

None

**Other Strengths And Weaknesses:**

Strengths:
-OMG delineates how to perform CSP and DNG with the stochastic interpolants framework and provides informative ablations with SOTA results.

-The use of DFM for atomic species generation and DNG is novel to my knowledge.

-The open-source implementation contributes to reproducibility and benchmarking for future works.

Weaknesses:

-The claims of unification of other material generation approaches under SI is not completely supported, although SI does offer some additional tuning knobs over basic CFM probability paths.

-The ideas here are mostly extensions of previous formulations and not incredibly novel.

**Questions For Authors:**

1. To support this claim of “unification” can we show explicitly how each previous framework is realized by SI? It could be conceptually and practically interesting to catalog the previous approaches. This would make it clear how one would propose future extensions of those methods under the SI framework.

2. Could the periodic interpolants and how they are supported by the SI framework be formulated in more detail? I'm not very convinced by the section 4.1 and appendix regarding the well-definedness of the paths and their geodesics on flat tori. This would help us better contextualize this framework with what FlowMM is doing with Riemannian flow-matching and propose extensions.

3. Would it be possible to compare any computational overheads for training and sampling across SI and the related methods? This would further help us understand tradeoffs and address any issues regarding scalability.

These points would make me feel more confident about the contributions of this work.

**Relation To Broader Scientific Literature:**

I find it pretty clear to follow OMG’s relationship to previous works DiffCSP and FlowMM in the setting of CSP and DNG tasks.

**Theoretical Claims:**

It is suggested that SI generalizes both diffusion and flow-matching frameworks, but the equivalence to previous frameworks isnt explicitly noted anywhere. The authors mention the use of periodic interpolants to account for PCBs which seems to be compatible with the original SI framework, but it seems the paper mostly gives intuitive arguments for this, citing Albergo 2023 and Jiao 2023.

---

> ### Author Rebuttal · Authors · 2025-04-01
>
> We thank the reviewer for the thoughtful and constructive comments. Below we address the main concerns and the three questions raised.
> ## Novelty and contribution
> We acknowledge that the novelty of our work could have been more clearly emphasized. To clarify:
> - While DiffCSP and MatterGen specifically implement score-based diffusion with SDE-based sampling, and FlowMM uses a fixed linear interpolant with ODEs, our approach in OMG builds on the broader **stochastic interpolants (SI) framework** which enables **both ODE- and SDE-based generation and a much wider range of interpolants**. To our knowledge, this flexibility has not previously been explored for crystal generation. By systematically studying this much broader design space, we demonstrate **state-of-the-art performance** across the CSP and DNG tasks. We also set the first CSP baseline for the Alex-MP dataset by reporting OMG's performance on it.
> - We **refine the match-rate metric for CSP** by eliminating unnecessary filtering present in prior work (e.g., CDVAE, DiffCSP and FlowMM), and we **introduce the average coordination number metric for DNG** to better evaluate the similarity of generated and test structures.
> - From a methodology perspective, OMG is the **first work to incorporate periodic boundary conditions (PBCs)** into the SI framework. As noted by the reviewer, the use of **discrete flow matching (DFM) for atomic species generation in DNG** is also novel.
> - We introduce the **minimum permutation distance** option as a data-dependent coupling during training that permutes atoms within structures to minimize the per-atom displacement during interpolation.
> - Our proposed framework is highly flexible and extensible. It can be **easily adapted for LLM-enhanced material generation** (see response to Reviewer w22y for OMG-LLM / FlowLLM results).
> ## Claim of unification
> We thank the reviewer for raising this point and agree it would strengthen the paper to make the unification claim more explicit. **We will add a dedicated section in the appendix** cataloging how prior approaches can be recovered within the SI framework:
> - **Conditional Flow Matching (CFM)** as implemented in FlowMM is naturally subsumed by SI. When using ODE-based sampling [considering only the loss in Eq. (2)] with the linear interpolant $x(t, x_0, x_1) = (1 - t) x_0 + t x_1$, Eq. (2) becomes identical to the FlowMM loss [see Eq. (15) in Miller et al., 2024; arXiv:2406.04713].
> - **Score-based diffusion models (SBDMs)** are recovered via specific stochastic interpolants, both in their **variance-preserving** (VP) and **variance-exploding** (VE) forms as they appear in DiffCSP and MatterGen (see Aranguri et al., 2025; arXiv:2501.00988). The SBD interpolant $x(t, x_0, x_1) = \sqrt{1 - t^2} x_0 + t x_1$ (derived in Section 5.1 of Albergo et al., 2023; arXiv:2303.08797) recovers the VP variant of SBDMs. In this work we only explicitly employ this SBD interpolant, as we only implemented the spatially linear interpolants outlined in Section 4 of (Albergo et al., 2023). Nevertheless, we will report the appropriate choices of interpolants to recover both VP and VE SBDMs in the revised discussion.
> ## PBCs
> We agree that the discussion of periodic boundary conditions (PBCs) should be extended:
> - We do not attempt to generalize stochastic interpolants (SIs) to arbitrary manifolds (as in Riemannian flow matching, or RFM). Instead, we adopt a task-specific formulation tailored to flat tori, which are the relevant manifolds for fractional coordinates in crystal generation.
> - As in FlowMM, in order to uniquely define the interpolating paths, we rely on shortest geodesic interpolation paths between pairs of fractional coordinates $x_0$ and $x_1$, ensuring that interpolants are well-defined and differentiable. As briefly discussed in Section 3.2.1, this shortest geodesic path can be computed by first _unwrapping_ one of the coordinates (say $x_1$) into its periodic image $x_1^{\prime}$, such that it is the closest image to $x_0$. We then compute the linear interpolant $x(t, x_0, x_1^{\prime}) = (1 - t) x_0 + t x_1^{\prime}$ as if in Euclidean space, and finally wrap the result back onto the torus. This yields exactly the same shortest-path geodesic as in FlowMM, and thus recovers its corresponding CFM loss.
> - All periodic stochastic interpolants are then defined similarly by computing $x(t, x_0, x_1^{\prime}, z) = \alpha(t) x_0 + \beta(t) x_1^{\prime} + \gamma(t) z$ in the unwrapped (Euclidean) space and wrapping back onto the torus. In Appendix A.3, we show that averaging over the latent variable $\gamma(t) z$ recovers the deterministic base interpolant path, as required by the SI framework.
>
> **We will revise and expand our discussion in Section 3.2.1 and Appendix A.3 to elaborate on this approach**, clarify its compatibility with SI, and contrast it more explicitly with RFM.
> ## Computational overheads
> We agree this is an important point. **We refer to our response to Reviewer w22y.**

---

### Decision · Program_Chairs · 2025-05-01

**Decision:**

Accept (poster)

**Comment:**

This paper presents the method referred to as _Open Materials Generation (OMG) with Stochastic Interpolants (SI)_. The method is designed to tackle the problem of generating crystal structures and the paper offers an evaluation on _de novo_ crystal structure generation and crystal structure prediction from compositions.

The reviews are overall positive, with a consensus in the recommendation scores of _Weak accept_. The reviewers seem enthusiastic about the positive results and do not critique the claims of achieving state of the art performance. My own assessment and critique is that the metrics used in the evaluation are a poor indication of the usefulness in practical downstream applications in materials discovery. Nonetheless, this is an unfortunate trend in this area rather than a unique flaw of this paper.

The main criticisms are 1) that the comparisons with other methods are not apples-to-apples comparisons and 2) that the claims of unifying FlowMM and diffusion models are not fully supported. In my view, the rebuttal responses are not solid in addressing these comments and I would strongly encourage the authors to improve these aspects in a subsequent update of the paper.

Despite these weaknesses and the lack of more enthusiastic comments by the reviewers, my position is to not recommend the rejection of papers that have sufficient scientific quality, are sufficiently well-written and that could be interesting to part of the ICML audience. This paper should pass that bar.